# Canonical NF-κB signaling maintains corneal epithelial integrity and prevents corneal aging via retinoic acid

**Qian Yu[1], Soma Biswas[2], Gang Ma[1], Peiquan Zhao[2], Baojie Li[1,3]\*, Jing Li[2]\***

[1]Bio-X Institutes, Key Laboratory for the Genetics of Developmental and Neuropsychiatric Disorders, Ministry of Education, Shanghai Jiao Tong University, Shanghai, China; [2]Department of Ophthalmology, Xinhua Hospital affiliated to Shanghai Jiao Tong University School of Medicine, Shanghai, China; [3]Institute of Traditional Chinese Medicine and Stem Cell Research, School of Basic Medicine, Chengdu University of Traditional Chinese Medicine, Chengdu, China

**Abstract** Disorders of the transparent cornea affect millions of people worldwide. However, how to maintain and/or regenerate this organ remains unclear. Here, we show that *Rela* (encoding a canonical NF-κB subunit) ablation in K14[+] corneal epithelial stem cells not only disrupts corneal regeneration but also results in age-dependent epithelial deterioration, which triggers aberrant wound-healing processes including stromal remodeling, neovascularization, epithelial metaplasia, and plaque formation at the central cornea. These anomalies are largely recapitulated in normal mice that age naturally. Mechanistically, *Rela* deletion suppresses expression of Aldh1a1, an enzyme required for retinoic acid synthesis from vitamin A. Retinoic acid administration blocks development of ocular anomalies in *Krt14-Cre; Rela[f/f]* mice and naturally aged mice. Moreover, epithelial metaplasia and plaque formation are preventable by inhibition of angiogenesis. This study thus uncovers the major mechanisms governing corneal maintenance, regeneration, and aging and identifies the NF-κB-retinoic acid pathway as a therapeutic target for corneal disorders.

**\*For correspondence:**
libj@sjtu.edu.cn (BL);
lijing@xinhuamed.com.cn (JL)

**Competing interests:** The authors declare that no competing interests exist.

## Introduction

The cornea is a transparent organ composed of the epithelial, stromal, and endothelial layers (*Yazdanpanah et al., 2017*; *Nowell and Radtke, 2017*; *Collinson et al., 2004*), with the epithelial basement membrane separating the epithelial and the stroma layers and the Descemet's basement membrane separating the stroma and endothelial layers (*Wilson, 2020*; *Medeiros et al., 2018*). While the epithelial layer undergoes constant turnover driven by corneal epithelial stem cells (CESCs), the stromal and endothelial layers are relatively quiescent in adults (*Castro-Muñozledo, 2013*; *Majo et al., 2008*; *Schlötzer-Schrehardt and Kruse, 2005*). Corneal transparency is attributable to the lack of keratin secretion by epithelial cells, the specific arrangement of stromal lamellae, expression of crystallins by keratocytes, and the lack of vasculature (*Nowell and Radtke, 2017*; *Jester, 2008*; *Yam et al., 2020*). The cornea contributes 65–75% of the eye's total focusing power and provides a barrier to protect against environmental insults (*Yazdanpanah et al., 2017*). Thus, a clear and healthy cornea is a prerequisite for proper vision.

As the outermost part of the eye, the cornea is often challenged by external stimuli, including UV radiation, chemicals, traumatic abrasions, or bacterial infection (*Nowell and Radtke, 2017*; *Lyu et al., 2020*). In addition, contact lens usage and eye surgery can also cause corneal injuries (*Toda, 2008*). The damaged corneal epithelial layer is replenished by CESCs, which are mainly located in the limbal region in humans and genetically identified by ABCB2, ABCB5, keratin5/14, or p63 expression in mice (*Schlötzer-Schrehardt and Kruse, 2005*; *Notara et al., 2010*;

*Cotsarelis et al., 1989*; *Di Iorio et al., 2005*; *Ksander et al., 2014*). Corneal repair often involves inflammation and activation of keratocytes and improper repair may cause epithelial degeneration and dystrophy, scar formation, neovascularization, and the loss of transparency, which affect millions of people worldwide, eventually necessitating corneal transplantation, which is limited by a lack of corneal donors (*Wong et al., 2017*; *Ljubimov and Saghizadeh, 2015*).

Several studies have revealed cross-talks between the stromal and epithelial layers during corneal development and regeneration (*AbuSamra et al., 2019*; *Dziasko and Daniels, 2016*). Growth factors secreted by keratocytes, particularly Wnt and EGF, promote epithelial cell proliferation via transcription factors such as β-catenin and Pax6 (*Nakatsu et al., 2011*; *Ouyang et al., 2014*; *McClintock and Ceresa, 2010*; *Kumar and Duester, 2010*). In addition, keratocytes and immune cells also secrete cytokines, such as IL6, to promote epithelial cell proliferation during repair (*Notara et al., 2010*). On the other hand, corneal epithelial cells secrete fibroblast growth factors (FGF) to promote stromal cell proliferation (*Vauclair et al., 2007*). Still, how corneal epithelial and stromal layers interact under pathological conditions is not well understood.

Many environmental insults to the cornea are potential activators of NF-κB, which is a central player in the inflammatory response and cell proliferation and survival (*Taniguchi and Karin, 2018*; *Eluard et al., 2020*). Previous studies have reported NF-κB activation in corneal epithelial cells, macrophages, and keratocytes, which is required for cytokine production, corneal regeneration, and fibrosis (*Zhou et al., 2021*; *Nakano et al., 2018*; *Lennikov et al., 2018*; *Orita et al., 2013*; *Oh et al., 2012*; *Chen et al., 2016*). NF-κB activation may promote corneal healing via upregulating the expression of CCCTC-binding factors in epithelial cells (*Wang et al., 2013*). To further understand the functions of the NF-κB pathway in corneal homeostasis and regeneration, we ablated *Rela*, which encodes a subunit of canonical NF-κB signaling complex, in K14+ CESCs or keratocytes. While *Rela* ablation in keratocytes did not affect corneal homeostasis or regeneration, *Krt14-Cre; Rela^{f/f}* mice showed age-dependent deterioration in the epithelial layer at the central cornea, which triggered an aberrant repair response that led to inflammation, neovascularization, epithelial metaplasia, and plaque formation at the central cornea. Interestingly, these corneal phenotypes were largely recapitulated in naturally aged mice, associated with decreased NF-κB activation and RelA expression. Mechanistically, NF-κB positively regulated *Aldh1a1* transcription and retinoic acid (RA) synthesis from vitamin A and that the development of corneal phenotypes in *Rela*-deficient mice or naturally aged mice was largely prevented by RA administration. While vitamin A deficiency is known to cause night blindness, corneal ulcers, and eye development defects, mainly in children (*Smith et al., 2018*; *Fares-Taie et al., 2013*; *Srour et al., 2013*; *Yahyavi et al., 2013*; *Verhoeven et al., 2013*), we show for the first time that RA, the metabolic product of vitamin A, is critical in corneal regeneration and aging in mice, implying that RA has the potential to improve corneal health in adult and aged individuals.

## Results

### *Rela* ablation in K14+ CESCs, but not stromal cells, impairs corneal regeneration

The ocular surface is constantly exposed to environmental stimuli, many of which are NF-κB activators (*Eluard et al., 2020*). Indeed, western blot analysis revealed activation of NF-κB in the corneal tissues of standard housed mice, as indicated by the presence of p-RelA and p-IKKα (*Figure 1A*). Immunohistochemical staining confirmed nuclear accumulation of RelA in the central region but not much in peripheral or limbal epithelial cells (*Figure 1B*). Activation of NF-κB signaling was also observed in corneal stromal cells, although to a reduced extent (*Figure 1B*). NF-κB activation was increased in both epithelial and stromal cells after alkaline burn (*Figure 1A, B* and *Figure 1—figure supplement 1*). This finding is consistent with the notion that the basal activity of NF-κB is low but activated during wound healing (*Zhang et al., 2017*) and suggests that NF-κB is involved in corneal regeneration.

We then crossed *Rela^{f/f}* and *Krt14-Cre* mice to generate *Krt14-Cre; Rela^{f/f}* mice to study the functions of canonical NF-κB signaling in corneal epithelial homeostasis. Consistent with previous studies (*Zhao et al., 2008*), lineage tracing with *Krt14-Cre; ROSA26^{fs-tdTomato}* mice showed Tomato+ labeling of corneal epithelial cells, limbal CESCs, conjunctiva, and meibomian glands, as well as the skin,

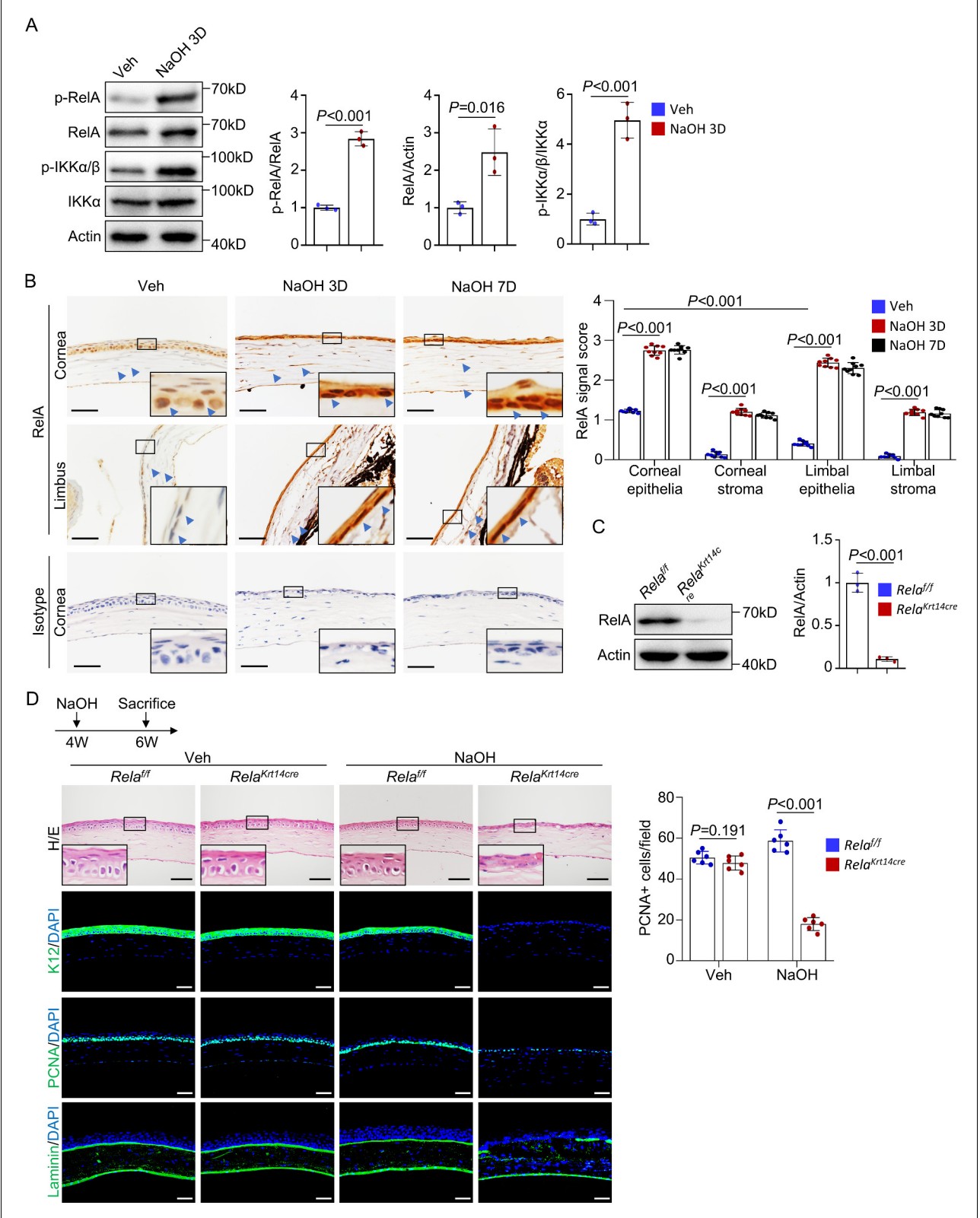

**Figure 1.** *Rela* ablation in K14[+]corneal epithelial stem cells impaired corneal regeneration. (**A**) Representative western blot results showed enhanced activation of NF-κB in corneal samples during regeneration. The blots were probed with antibodies against RelA, p-RelA, IKKα, or p-IKKα/β. Right panel: quantitation results. n = 3 per group. (**B**) Representative immunohistochemical staining results showed nuclear localization of RelA in epithelial and stromal cells in normal and regenerating corneas. Arrowheads: RelA signals. Scale bar, 50 μm. Right panel: quantitation results. n = 3 views/sample

*Figure 1 continued on next page*

Figure 1 continued

× 3 samples per group. (C) Western blot results showed that RelA level was drastically reduced in *Krt14-Cre; Rela^f/f* mouse corneal samples. Right panel: quantitation results. n = 3 per group. (D) Histological analyses revealed that cornea repair was defective in *Krt14-Cre; Rela^f/f* mice. The cornea sections were stained with H/E or antibodies against PCNA, K12, or laminin. Scale bar, 50 μm. Upper panel: diagram showing the time of injury and mouse euthanization. Right panel: quantitation results. n = 6 per group. Data was presented as mean ± SEM. Unpaired two-tailed Student's t-test was applied for (A, C), and two-way ANOVA was applied in (B, D). p-value<0.05 was considered as statistically significant.

The online version of this article includes the following source data and figure supplement(s) for figure 1:

**Source data 1.** Numeric data used in *Figure 1*.
**Figure supplement 1.** The corneal regeneration process after alkaline burn.
**Figure supplement 2.** Tracing results of *Krt14-Cre; ROSA26^fs-tdTomato* mice.
**Figure supplement 3.** Deletion of *Rela* in stromal cells did not affect corneal homeostasis or regeneration.

oral mucosa, and trachea but not intestinal epithelial cells (*Figure 1—figure supplement 2A, B*). The corneas of young *Krt14-Cre; Rela^f/f* mice appeared normal up to 6 weeks of age (*Figure 1C , D*), suggesting that the canonical NF-κB pathway is dispensable for corneal epithelial development and early postnatal growth. In the mutant mice, expression of K14, a CESC marker, was not affected in central or peripheral cornea or the limbus (*Figure 1—figure supplement 2C*), suggesting that the genetic manipulation of the *Krt14* or *Rela* locus or *Rela* ablation does not affect K14 expression.

When 4-week-old *Krt14-Cre; Rela^f/f* mice were subjected to alkaline burn on the ocular surface, severe defects in corneal wound healing were observed (*Figure 1D*). We used 4-week-old mice to exclude possible effects of corneal growth defects on regeneration as the corneas of *Krt14-Cre; Rela^f/f* mice started to show anomalies at 6 weeks of age. Unlike the controls, the regenerated corneal epithelium lacked a basal layer and epithelial basement membrane (*Figure 1D*). Immunostaining revealed the absence of differentiation marker K12, decreased numbers of PCNA^+ proliferating cells, and disrupted Laminin^+ basement membrane at the injury sites, although these anomalies were not observed in the peripheral or limbal epithelia (*Figure 1D* and *Figure 1—figure supplement 2D*). Overall, these results indicate that *Rela* ablation impairs central corneal epithelial cell proliferation and differentiation during regeneration.

To study the function of RelA in keratocytes, we generated *Prrx1-Cre; Rela^f/f* mice. Prrx1 is a marker for mesenchymal stromal cells, and the *Prrx1-Cre* mice have been widely used to study skeletal development (*Logan et al., 2002*). Our genetic tracing with *Prrx1-Cre; ROSA26^fs-tdTomato* mice revealed that Prrx1 marked most corneal stromal cells (*Figure 1—figure supplement 3A*). *Prrx1-Cre; Rela^f/f* mice had a normal corneal structure, and the healing process after alkaline burn injury was normal compared to that of controls (*Figure 1—figure supplement 3B, C*). A previous study showed that the deletion of *Ikkb*, which encodes a kinase required for activation of both canonical and noncanonical NF-κB pathways, in keratocytes (via *Keratocan-Cre*) impaired corneal repair but not maintenance (*Chen et al., 2016*). These results suggest that the non-canonical rather than the canonical NF-κB pathway in stromal cells plays a role in corneal regeneration.

### *Rela* ablation causes age-dependent epithelial deterioration at the central cornea

Further analysis of *Krt14-Cre; Rela^f/f* mice revealed age-dependent changes in corneal structures, including visible plaques at the central cornea by 6 months of age (*Figure 2A*). Histological analysis showed that the mutant mice had largely normal corneas up to 1.5 months of age. However, at 2 months of age, the mutant mice showed basal cell atrophy at the central cornea, and the degenerative changes expanded to suprabasal cells at the age of 2.5 months (*Figure 2B*). By 3 months of age, the central corneal epithelium had disappeared. In 2-month-old mutant mice, the central cornea had fewer PCNA^+ or p63^+ cells and showed decreased expression of K12 at the mRNA and protein levels (*Figure 2C–E*). However, the number of apoptotic cells, which were mainly detected at the central region, remained similar between the mutant and control mice (*Figure 2—figure supplement 1*). The integrity of the epithelial layer was disrupted as the corneal epithelial layer could be scraped off more easily in mutant mice than in control mice (*Figure 2F*). In addition, we observed drastic LC-biotin diffusion from outside the epithelial layer into the stroma in 2- but not 1-month-old mutant mice (*Figure 2G*).

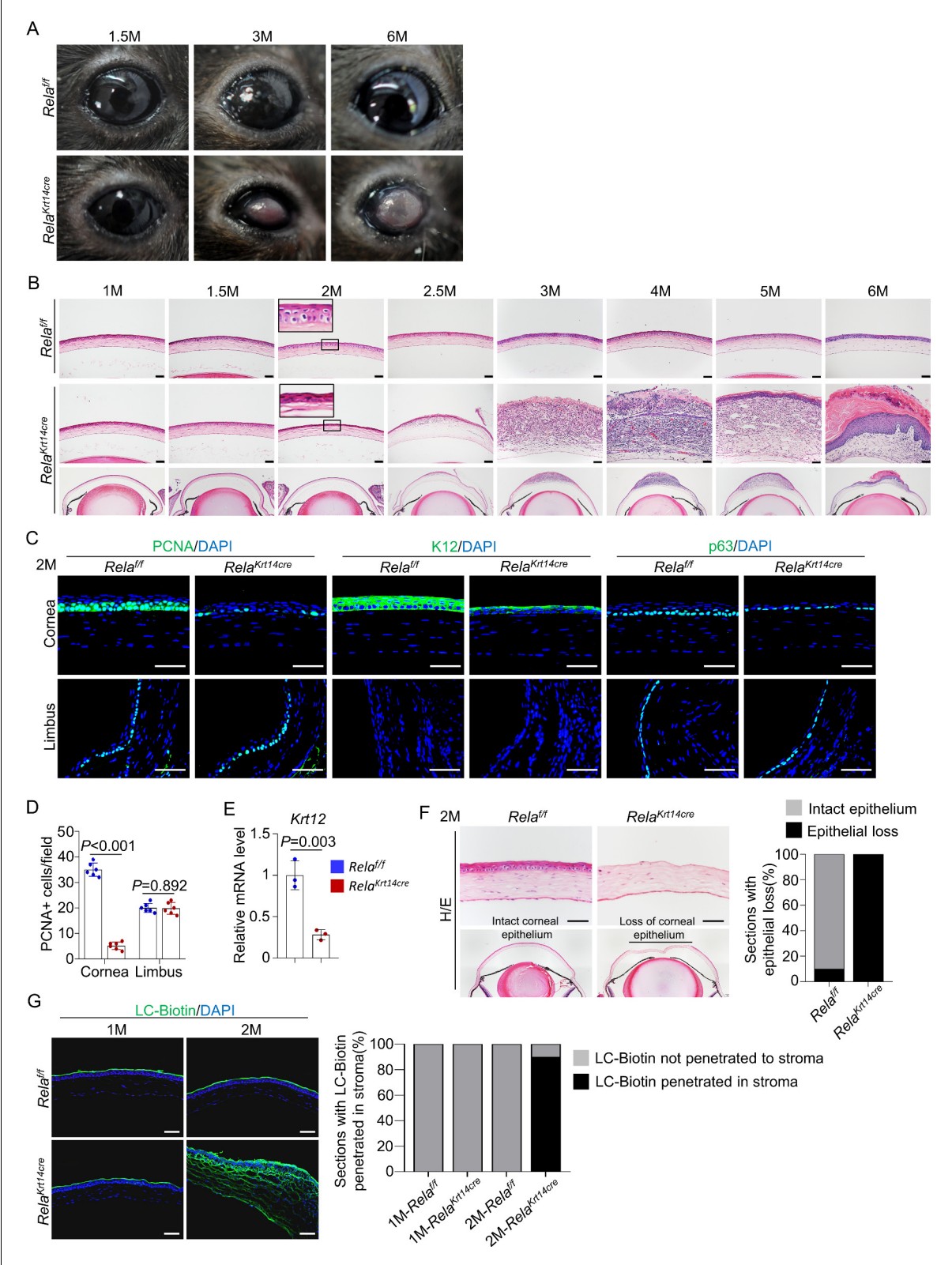

**Figure 2.** *Rela* ablation causes age-dependent epithelial deterioration and plaque formation at the central cornea. (**A**) Representative images showed the plaques formed at the central cornea of all 6-month-old *Krt14-Cre; Rela*^f/f^ mice. n = 10 per group. (**B**). Representative H/E staining results showed an age-dependent change in cornea structures of *Krt14-Cre; Rela*^f/f^ mice. Scale bar, 50 μm. (**C**). Representative immunostaining results showed that both epithelial proliferation and differentiation were defective in 2-month-old *Krt14-Cre; Rela*^f/f^ mice. Cornea sections were stained for PCNA, p63, or
*Figure 2 continued on next page*

*Figure 2 continued*

K12. Scale bar, 50 µm. (D). Quantitation data of PCNA$^+$ proliferating cells. n = 6 per group. (E). qPCR analysis showed that *K12* mRNA levels were decreased in 2-month-old *Krt14-Cre; Rela$^{f/f}$* mouse corneal epithelial samples compared to control samples. n = 3 per group. (F). Corneal fragility assay showed that corneal epithelial layer was scraped off more easily in mutant mice than control mice. Scale bar, 50 µm. Right panel: quantitation results. n = 10 per group. (G). LC-biotin staining assays showed that the integrity of the epithelial layer was disrupted in 2- but not 1-month-old *Krt14-Cre; Rela$^{f/f}$* mice. Scale bar, 50 µm. Right panel: quantitation results. n = 10 per group. Data was presented as mean ± SEM. Unpaired two-tailed Student's t-test was applied in (D, E). p-value<0.05 was considered as statistically significant.

The online version of this article includes the following source data and figure supplement(s) for figure 2:

**Source data 1.** Numeric data used in *Figure 2*.
**Figure supplement 1.** Corneal epithelia shown no alteration in apoptosis in *Krt14-Cre; Rela$^{f/f}$* mice.
**Figure supplement 1—source data 1.** Numeric data used in *Figure 2—figure supplement 1*.
**Figure supplement 2.** The peripheral and limbal regions appeared normal in *Krt14-Cre; Rela$^{f/f}$* mice.

Interestingly, the peripheral and limbal regions appeared unaltered in *Krt14-Cre; Rela$^{f/f}$* mice (*Figure 2—figure supplement 2*), and the numbers of PCNA$^+$ proliferating cells and p63$^+$ CESCs were similar to those of control mice (*Figure 2c*). Overall, these findings indicate that RelA is required for corneal epithelial cell proliferation and differentiation mainly at the central cornea under homeostatic and repair conditions. This is consistent with greater activation of NF-κB in the central cornea than the peripheral cornea and limbus (*Figure 1B*), which may be caused by constant exposure of the central cornea to external environment.

## Stromal remodeling and neovascularization follow epithelial deterioration

The changes in the central corneal epithelium were followed by remodeling of the stroma underneath the epithelial layer in the mutant mice (*Figure 2B*). Beginning at 3 months of age, we found keratocyte hyperproliferation, leukocyte infiltration, and generation of blood and lymphatic channels, which progressed with advancing age. Immunofluorescent staining further supported the histological results, with the observation of cells that stained positive for CD31, Lyve1, or CD45 in the stroma (*Figure 3A*). The stromal cells expressed high levels of vimentin (a fibroblast marker) but not αSMA (a myofibroblast and smooth muscle marker) (*Medeiros et al., 2018*). It also showed increased expression of fibroblast-specific protein 1 (FSP1) and increased Picrosirius red staining signals (*Figure 3A* and *Figure 3—figure supplement 1*). These results suggest that stroma expansion was not typical fibrosis in *Krt14-Cre; Rela$^{f/f}$* mice. Stromal remodeling has been observed in mouse models fed a vitamin A-deficient diet or with epithelial-specific *Notch1* ablation (*Toshino et al., 2005*; *Nowell et al., 2016*). In *Krt14-Cre; Rela$^{f/f}$* mice, stroma remodeling was likely secondary to deterioration of the epithelial layer as it only occurred after 3 months of age when the epithelial layer had been disrupted (*Figure 3—figure supplement 2*).

## Epithelial cells convert to epidermal cells and form plaques at the central cornea

Intriguingly, a new layer of epithelium formed at 5 months of age in the mutant mice. By 6 months, the central cornea was covered by a thick layer of keratin and epithelium with invagination into the underlying stroma (*Figure 2B*), which are features characteristic of hyperproliferative skin. This finding is consistent with our observation that more basal cells and cells immediately above the basal cells, but not limbal cells, were positive for PCNA (*Figure 3B*).

To examine the identity and features of the newly formed epithelium, we separated the epithelial layers of 6-month-old *Krt14-Cre; Rela$^{f/f}$* and age-matched control mice and performed RNA-seq analysis (*Figure 3C* and *Figure 3—figure supplement 3A, B*). We found that the most highly enriched genes in the mutant samples were epidermal-related genes, in addition to protein synthesis and cell cycle genes (*Figure 3D, E* and *Figure 3—figure supplement 3C*). Immunostaining confirmed the switch from the expression of corneal epithelium-specific K12 to epidermal-specific K1 and loss of Pax6 expression (*Figure 3F*), confirming the occurrence of metaplasia in corneal epithelial cells.

Next, we generated *Krt14-Cre; Rela$^{f/f}$; ROSA26$^{fs-tdTomato}$* mice. Tracing at 6 months of age revealed that the excessively proliferative dermal-like cells were all Tomato$^+$ (*Figure 3G*), indicating

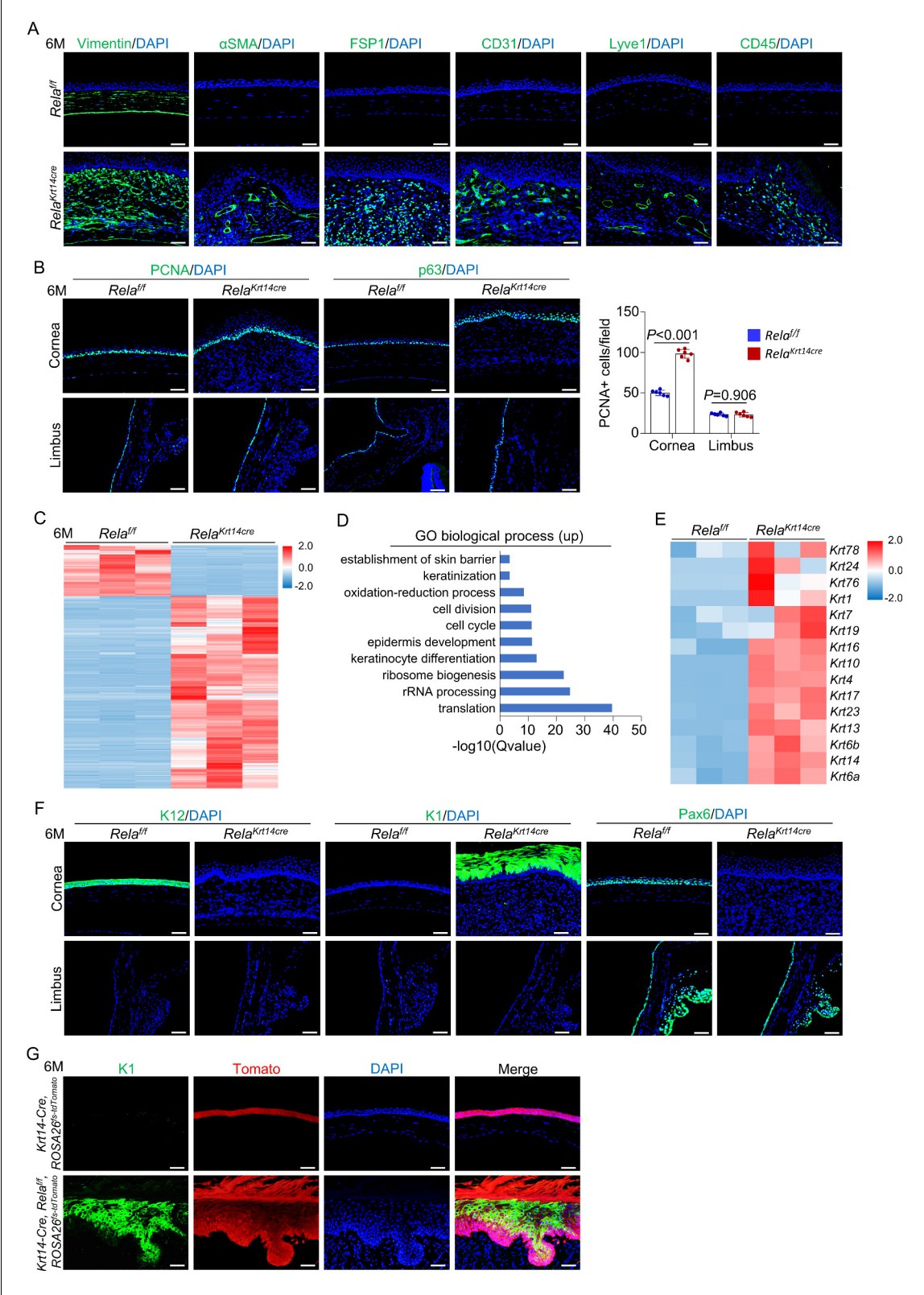

**Figure 3.** *S*troma remodeling, neovascularization, and metaplasia at central cornea in *Krt14-Cre; Rela^f/f* mice. (**A**) Representative immunostaining results showed that vimentin, αSMA, FSP1, CD31, Lyve1, and CD45 signals were increased in 6-month-old *Krt14-Cre; Rela^f/f* mice compared to control mice. Scale bar, 50 μm. (**B**) Representative immunostaining results showed that PCNA and p63 signals were increased at the central cornea but not the limbus in 6-month-old *Krt14-Cre; Rela^f/f* mice compared to control mice. Scale bar, 50 μm. Right panel: quantitation data. n = 6 per group. (**C**) Heatmap

*Figure 3 continued on next page*

*Figure 3 continued*

of top 2000 genes expressed in the corneal epithelia of 6-month-old *Krt14-Cre; Rela*^f/f^ mice compared to control mice. n = 3 per group. (**D**) GO biological process analysis of upregulated genes in the mutant samples. (**E**) Heatmap of keratin genes expressed in corneal epithelial samples of 6-month-old *Krt14-Cre; Rela*^f/f^ mice compared to control mice. n = 3 per group. (**F**) Representative immunostaining results for K12, K1, and Pax6 in the corneal epithelia of 6-month-old *Krt14-Cre; Rela*^f/f^ mice. Scale bar, 50 µm. (**G**) Tracing of Tomato⁺ cells revealed that K1-expressing dermal-like cells were derived from K14⁺ cells. Six-month-old *Krt14-Cre; Rela*^f/f^; *ROSA26*^fs-tdTomato^ mice were used. Scale bar, 50 µm. Data was presented as mean ± SEM. Unpaired two-tailed Student's t-test was applied in (**B**). p-value<0.05 was considered as statistically significant.

The online version of this article includes the following source data and figure supplement(s) for figure 3:

**Source data 1.** Numeric data used in *Figure 3*.
**Figure supplement 1.** *Krt14-Cre; Rela*^f/f^ mice showed fibrosis in the cornea.
**Figure supplement 2.** *Krt14-Cre; Rela*^f/f^ mice showed blood vessels formation and immune cell infiltration at 3 but not 2 months of age.
**Figure supplement 3.** RNA-seq results of corneal samples of 6-month-old *Krt14-Cre; Rela*^f/f^ and control mice.
**Figure supplement 3—source data 1.** Numeric data used in *Figure 3—figure supplement 3*.
**Figure supplement 4.** Other tissues marked by K14 were normal in 6-month-old *Krt14-Cre; Rela*^f/f^ mice.
**Figure supplement 4—source data 1.** Numeric data used in *Figure 3—figure supplement 4*.

that these cells were derived from either the remaining K14⁺ cells at the central cornea or K14⁺ cells at the peripheral cornea. Corneal cell metaplasia has been observed in mouse models of chronic inflammation, *Pax6*-deficient mice, or in mice fed a vitamin A-deficient diet and is believed to occur in stem cells or progenitor cells (*Ouyang et al., 2014*; *Vauclair et al., 2007*; *Nowell et al., 2016*; *Chen et al., 2009*). Collectively, these data suggest that epidermal-like cells originate from corneal epithelial progenitors, which undergo metaplastic changes in the central region in the presence of remodeling stroma in *Krt14-Cre; Rela*^f/f^ mice.

Although genetic tracing showed the presence of K14-labeled epithelial cells in the conjunctiva, meibomian glands, and epithelial cells of many other organs (*Figure 1—figure supplement 2A, B*), *Rela* ablation did not affect conjunctiva or meibomian glands, as well as the overall structure of the skin, oral mucosa, trachea, or intestine epithelia (*Figure 3—figure supplement 4A–C*). These results suggest that the effects of RelA are restricted to the central cornea, which may be caused by mitogenic signals secreted by the remodeling stroma, in cooperation with environmental cues.

## RelA is required for corneal epithelial proliferation and differentiation in vitro

To validate the roles of RelA in corneal epithelial cell proliferation and differentiation, we isolated corneal epithelial cells and briefly expanded them in proliferation medium (KSFM) before switching them to the differentiation medium. We found that *Rela*^-/-^ cells showed drastic defects in proliferation, as indicated by a reduction in the number of PCNA⁺ cells, and differentiation, as indicated by decreased expression of K12 at the mRNA and protein levels (*Figure 4—figure supplement 1A, B*). However, the epidermal markers K1 and K10 were not detectable in *Rela*^-/-^ or control cultures (*Figure 4—figure supplement 1C*). These results confirm the positive roles of RelA in corneal epithelial cell proliferation and differentiation and suggest that the subsequent epidermal conversion and plaque formation may not be cell autonomous. Rather, niche molecules provided by the stroma may play an important role in these events.

## *Rela* ablation suppresses *Aldh1a1* expression and RA synthesis

To further understand the mechanisms underlying *Rela* deletion-induced corneal defects, we isolated mRNA from the corneal epithelial cells of 2-month-old *Krt14-Cre; Rela*^f/f^ and control mice and performed bulk RNA-seq analysis. *Rela* ablation drastically affected the gene expression of corneal epithelial cells (*Figure 4A*). Gene ontology (GO) and kyoto encyclopedia of genes and genomes (KEGG) analyses revealed that *Rela* ablation led to a severe reduction in the expression of the RA synthesis enzyme Aldh1a1, as well as pathways that control cell proliferation, wound healing, and inflammation (*Figure 4B, C*), consistent with our in vivo results. RA is synthesized from vitamin A via two steps, and the first step is catalyzed by ADH/RDH, while the second step is catalyzed by Aldh1a1-3 (*Ghyselinck and Duester, 2019*). RA binds to RA receptors (RARs) and RA X receptors (RXRs) to control gene expression. RA is known to play important roles in corneal development and epithelial cell differentiation, and vitamin A deficiency causes night blindness, corneal ulcers, and

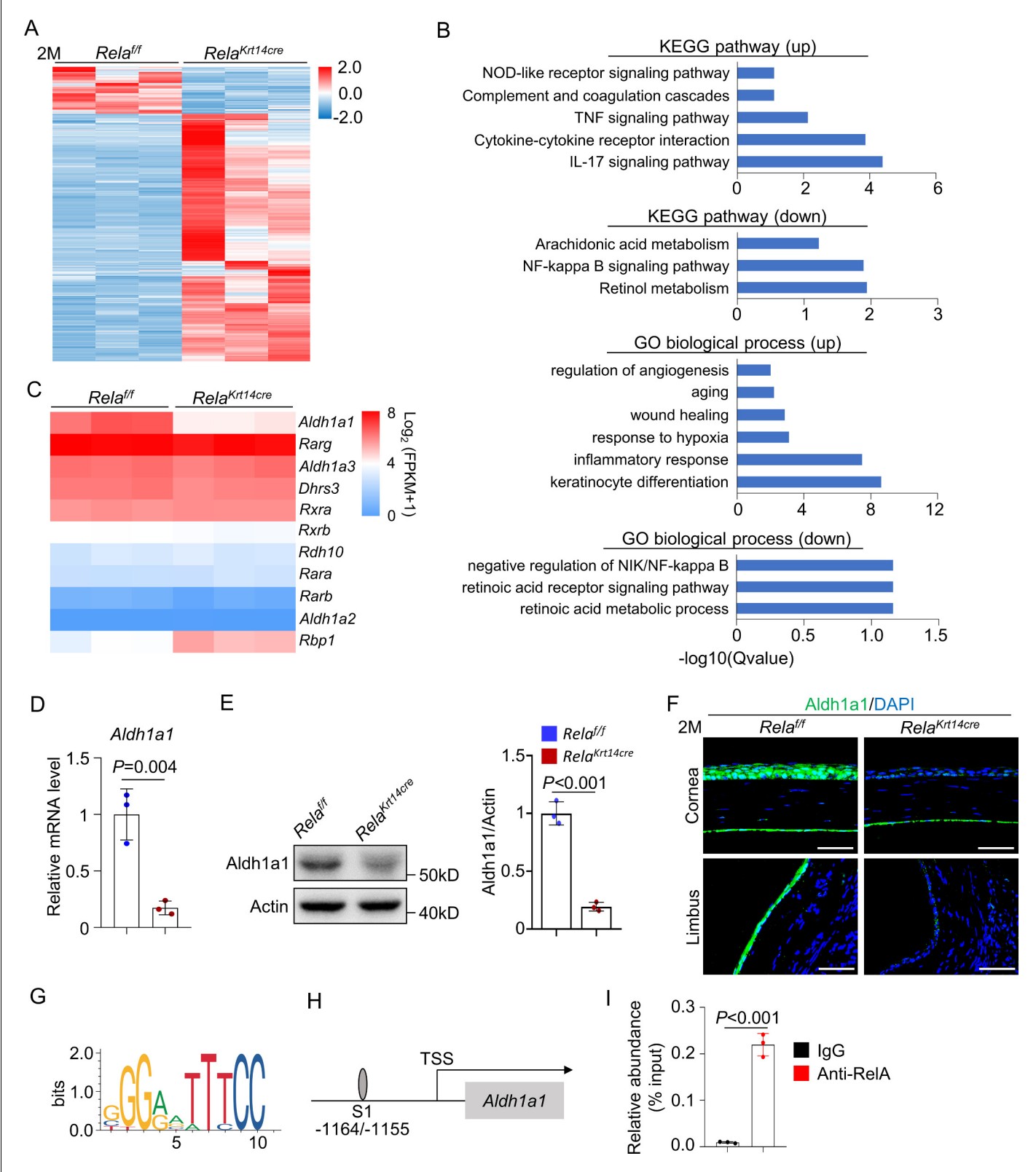

**Figure 4.** *Rela* ablation suppresses expression of retinoic acid synthesis enzyme Aldh1a1. (**A**) Heatmap of transcriptomes of corneal epithelial samples of 2-month-old *Krt14-Cre; Rela^f/f* and control mice. n = 3 per group. (**B**) KEGG analysis and GO analysis of corneal epithelial cells of 2-month-old *Krt14-Cre; Rela^f/f* and control mice. Both upregulated (up) and downregulated (down) pathways or modules were presented. (**C**) Heatmap of retinoic acid synthesis genes in corneal epithelial cells of 2-month-old *Krt14-Cre; Rela^f/f* mice compared to control mice. n = 3 per group. (**D**) Quantitative PCR
*Figure 4 continued on next page*

*Figure 4 continued*

showed downregulation of *Aldh1a1* expression in corneal epithelial samples of *Krt14-Cre; Rela^{f/f}* mice. n = 3 per group. (E) Representative western blot results showed that Aldh1a1 protein level was reduced in corneal samples of *Krt14-Cre; Rela^{f/f}* mice. Right panel: quantitation data. n = 3 per group. (F) Representative immunostaining results showed downregulation of Aldh1a1 in *Krt14-Cre; Rela^{f/f}* mouse cornea. Scale bar, 50 μm. (G) Sequence logos of 10-mer RelA-binding motif. (H) Schematic presentation of the putative RelA-binding site in the promoter region of *Aldh1a1*. (I) Quantitative PCR analysis of the immunoprecipitated DNA showed that *Aldh1a1* promoter had a RelA-binding site. n = 3 per group. Data was presented as mean ± SEM. Unpaired two-tailed Student's t-test was applied in (D, E, I). p-value<0.05 was considered as statistically significant.

The online version of this article includes the following source data and figure supplement(s) for figure 4:

**Source data 1.** Numeric data used in *Figure 4*.
**Figure supplement 1.** RelA was required for corneal epithelial cell proliferation and differentiation in vitro.
**Figure supplement 1—source data 1.** Numeric data used in *Figure 4—figure supplement 1*.
**Figure supplement 2.** Effects of *Rela* ablation on Aldha1 expression in conjunctiva and meibomian glands.

keratomalacia, mainly in children (*Kastner et al., 1994*; *Lassen et al., 2007*; *Kim et al., 2012*; *Macsai et al., 1998*). Whole-body deletion of *Aldh1a* genes causes xerophthalmia, as well as thinning of the stromal layer in young mice, which is followed by reduced proliferation and increased apoptosis of epithelial cells (*Kumar et al., 2017*). We confirmed that the mRNA levels *Aldh1a1* were decreased in *Rela*-deficient corneal samples (*Figure 4D*). Western blot and immunostaining confirmed decreased Aldh1a1 expression at the protein level in *Rela*-deficient samples (*Figure 4E, F*). These results suggest that RA synthesis is defective in the corneas of *Krt14-Cre; Rela^{f/f}* mice. Immunostaining revealed reduced expression of Aldh1a1 in the limbus and conjunctiva while the meibomian glands do not express Aldh1a1 (*Figure 4F* and *Figure 4—figure supplement 2*). The observation that the limbus and conjunctiva did not show obvious defects in *Krt14-Cre; Rela^{f/f}* mice indicates that the function of RA is also restricted to the central cornea.

We also confirmed the decrease in *Aldh1a1* mRNA in cultured *Rela^{-/-}* corneal epithelial cells compared to control cells (*Figure 4—figure supplement 1B*). Furthermore, we found that the promoter of *Aldh1a1* contained a putative NF-κB binding site (G/CGGA/GATTTCC) between −1164 and −1155 (*Figure 4G, H*). We then performed chromatin immunoprecipitation (ChIP) assays using anti-RelA antibodies in primary corneal epithelial cells. The ChIP results verified the presence of the NF-κB binding site in the *Aldh1a1* promoter (*Figure 4I*). Overall, these findings indicate that RelA directly regulates *Aldh1a1* expression in corneal epithelial cells.

## RA rescues *Rela* deletion-induced corneal homeostatic and regenerative defects

In cultured *Rela^{-/-}* corneal epithelial cells, we found that the defects in cell proliferation and differentiation were rescued by adding RA to the medium (*Figure 4—figure supplement 1A, B*). Moreover, in cultures of normal corneal epithelial cells, addition of an RA receptor inhibitor impaired cell proliferation and differentiation (*Figure 4—figure supplement 1D, E*). These results indicate that RA mediates the effect of *Rela* deletion on corneal epithelial cell proliferation and differentiation.

To determine whether RA rescues *Rela* ablation-induced corneal phenotypes in vivo, we treated *Krt14-Cre; Rela^{f/f}* and control mice with RA for 4.5 months starting at 1.5 months of age. Macroscopically, no corneal plaques were seen in RA-treated mice (*Figure 5A*). Histological and immunological analyses revealed that 6-month-old RA-treated mutant mice showed normal corneal epithelial and stromal structures (*Figure 5B–D*). In addition, 1.5 months of RA treatment, starting at 1.5 months of age, prevented thinning and dissolution of the epithelial layer at the central cornea (*Figure 5—figure supplement 1A*). These results suggest that a reduction in RA synthesis mediates the defects in the central cornea caused by *Rela* ablation. Moreover, when we administered RA to 4-week-old *Krt14-Cre; Rela^{f/f}* and control mice with alkaline-induced injury, RA-treated mutant mice were able to regenerate the epithelial layer as effectively as normal mice (*Figure 5E, F*). Note that the peripheral cornea and limbus were not affected by the injury (*Figure 5—figure supplement 1B*). Collectively, these findings indicate that canonical NF-κB signaling regulates corneal epithelial homeostasis and regeneration by promoting RA synthesis.

However, treatment of *Krt14-Cre; Rela^{f/f}* with RA at 3 months of age, after epithelial layer disruption, failed to prevent central corneal epithelial cell metaplasia, stromal remodeling, and plaque development (*Figure 5—figure supplement 1C*), indicating that RA could not reverse corneal

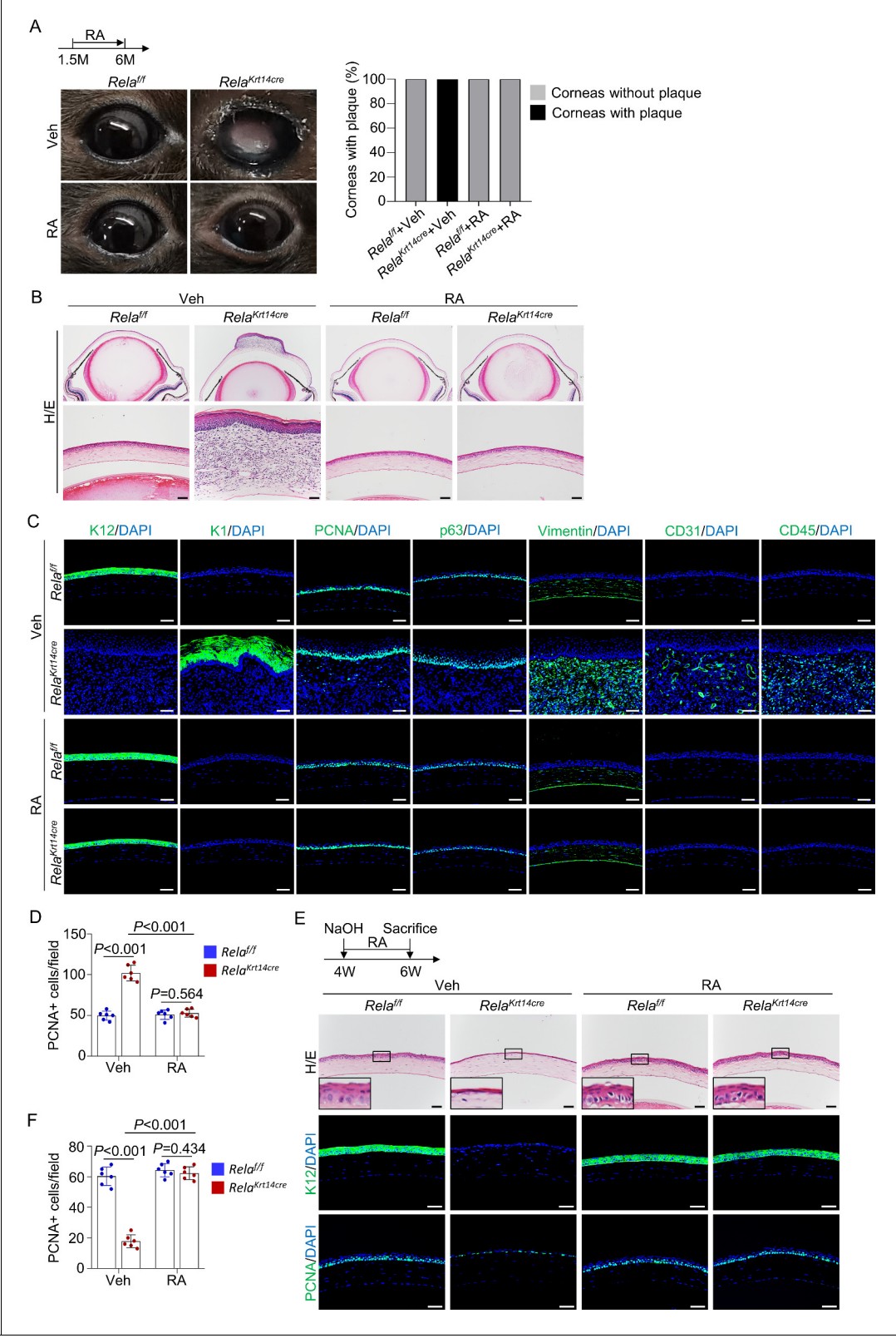

**Figure 5.** Retinoic acid (RA) diminishes homeostatic and regenerative defects caused by *Rela* ablation. (**A**) Representative images showed that RA blocked *Rela* ablation-induced plaque formation in *Krt14-Cre; Rela*^f/f mice. Upper panel: diagram showing the time of RA administration and mouse euthanization. Right panel: quantitation data. n = 10 per group. (**B**) Representative H/E staining results showed restoration of corneal structures by RA in *Krt14-Cre; Rela*^f/f mice. Scale bar, 50 µm. (**C**) Representative immunostaining results showed that epidermal fate conversion (K1 and K12), *Figure 5 continued on next page*

Figure 5 continued

overproliferation (PCNA and p63), stromal remodeling (vimentin), angiogenesis (CD31), and leukocyte infiltration (CD45) were all diminished by RA administration. Scale bar, 50 µm. (D) The percentage of PCNA$^+$ proliferating cells. n = 6 per group. (E) Representative histological results showed that RA rescued corneal regeneration defects in *Krt14-Cre; Rela*$^{f/f}$ mice. Upper panel: diagram showing the time of injury and RA administration. (F) The percentage of PCNA$^+$ proliferating cells. n = 6 per group. Data was presented as mean ± SEM. Two-way ANOVA was applied in (D, F). p-value<0.05 was considered as statistically significant.

The online version of this article includes the following source data and figure supplement(s) for figure 5:

**Source data 1.** Numeric data used in *Figure 5*.
**Figure supplement 1.** Retinoic acid (RA) rescued early epithelial defects but not stroma remodeling or epithelia metaplasia.
**Figure supplement 2.** Axitinib alleviates epidermal metaplasia and plaque formation in *Krt14-Cre; Rela*$^{f/f}$ mice.
**Figure supplement 2—source data 1.** Numeric data used in *Figure 5—figure supplement 2*.
**Figure supplement 3.** Enhanced activation of EGFR, Erks, and Stat3 in the metaplastic corneal tissues.
**Figure supplement 4.** Antibiotic levofloxacin did not rescue *Rela* ablation-induced corneal phenotypes.

epithelial metaplasia. Overall, these findings suggest that the primary epithelial defects (deterioration) caused by *Rela* deletion are attributable to a reduction in RA synthesis, whereas epithelial metaplasia and plaque formation are not directly caused by RA deficiency.

## The VEGFR inhibitor axitinib alleviates angiogenesis and epidermal metaplasia

Consistent with new blood vessel formation, the stroma showed an increase in expression of *Vegfa* but not *Vegfb*, *Vegfc*, or *Vegfd* in 3-month-old *Krt14-Cre; Rela*$^{f/f}$ mice (*Figure 5—figure supplement 2A*). VEGFA may underlie neovascularization and increased proliferation of corneal epithelial cells. To test this, we treated the mutant and control mice with VEGFR inhibitor axitinib, an approved drug for treating renal cell carcinoma that has been studied in the treatment of other cancer types (*Motzer et al., 2019*), for 4.5 months starting at 1.5 months of age. Axitinib not only prevented corneal angiogenesis but also suppressed epithelial cell overproliferation and metaplasia (*Figure 5—figure supplement 2B*), confirming the importance of VEGFA-VEGFR in these events. However, the central corneal epithelial layer remained disrupted, accompanied by decreased cell proliferation, with the epithelial layer being similar to that of 3-month-old *Krt14-Cre; Rela*$^{f/f}$ mice without treatment (*Figure 5—figure supplement 2B* and *Figure 2B*). The results suggested that axitinib did not affect development of early corneal epithelial defects caused by *RelA* ablation.

Consistent with hyperproliferation of metaplastic cells, western blot analysis revealed enhanced activation of the mitogenic Erk and Stat3 pathways but not Akt1 or β-catenin in corneal samples (*Figure 5—figure supplement 3A*). While VEGFA secreted by the stroma may contribute to activation of some of these signaling molecules, we found enhanced phosphorylation of EGFRs in the skin-like epithelial tissue (*Figure 5—figure supplement 3A*), which has an established role in corneal epithelial cell proliferation (*McClintock and Ceresa, 2010*). Immunostaining confirmed activation of EGFR, as well as Erk and Stat3 in metaplastic cells (*Figure 5—figure supplement 3B*). EGFs might be synthesized by remodeling stromal cells or brought in by newly formed blood vessels via blood circulation as it is well known that plasma contains high levels of EGF molecules (*Sánchez-Vizcaíno et al., 2010*).

## Antibiotic treatment shows little effect on corneal phenotypes

NF-κB is an important inflammatory signaling molecule, and chronic inflammation has been shown to induce corneal stromal remodeling and epithelial metaplasia (*Nowell et al., 2016*). We then examined whether bacterial infection played a role in the full spectrum of changes in the corneas of *Krt14-Cre; Rela*$^{f/f}$ mice as the mice showed open wounds in the cornea (*Figure 2B*). We treated mutant and control mice with levofloxacin, an antibiotic used to kill various types of bacteria, for 4.5 months starting at 1.5 months of age. While immune cell infiltration was largely blocked, stroma remodeling, angiogenesis, epidermal fate conversion, and plaque formation were not obviously affected (*Figure 5—figure supplement 4A, B*). Interestingly, we found that levofloxacin treatment did not affect NF-κB activation during regeneration in normal mice (*Figure 5—figure supplement 4C*). These results indicate that bacterial infection plays a minimal role in corneal defects of *Krt14-Cre; Rela*$^{f/f}$ mice.

## Natural aging induces corneal phenotypes similar to those of *Krt14-Cre; Rela^{f/f}* mice

The above results revealed age-dependent corneal anomalies in *Krt14-Cre; Rela^{f/f}* mice. Interestingly, thinning of the central corneal epithelium and skin-like keratinization were also observed in naturally aged mice, which had the same genetic background and were maintained in the same animal facility as *Krt14-Cre; Rela^{f/f}* mice. Approximately 12% of 24-month-old normal mice showed thinning of the corneal epithelial layer (hereafter designated moderate anomaly), 28% of mice showed epithelial overgrowth and plaque formation at the central cornea (hereafter designated severe anomaly), although 20-month-old mice showed no changes (*Figure 6A, B*). The observation that only a portion of aged mice showed corneal anomalies suggests the involvement of environmental factors in corneal aging. It is possible that thinning of the epithelial layer precedes plaque formation in aged mice. The thin corneal epithelium of 24-month-old mice showed a decrease in the numbers of proliferating cells and p63$^+$ CESCs and decreased expression of K12 and Pax6 but no change in apoptotic cells (*Figure 6C, D* and *Figure 6—figure supplement 1*). In corneas with plaque formation, the epithelial cells were K1-positive, and the thickened stroma was positive for FSP1 and vimentin, accompanied by neovascularization, lymphatic vessel formation, and immune cell infiltration (*Figure 6D*). However, the peripheral cornea and limbus region appeared normal in both groups of mice (*Figure 6C* and *Figure 6—figure supplement 2*). Similar changes in corneal thickness and corneal epithelial keratinization have been observed in aged humans, although they do not progress to plaque formation or blindness (*Yang et al., 2014*; *Blackburn et al., 2019*).

## RA prevents the development of aging-like corneal phenotypes in normal mice

We found that in the thin corneal samples of 24-month-old mice expression of RelA and Aldh1a1 and activation of NF-κB were significantly reduced compared to those of corneal samples from young mice (*Figure 7A*). qPCR analysis confirmed reduced expression of *Rela* and *Aldh1a1* at the mRNA level (*Figure 7B*). Immunohistochemical staining of RelA and immunostaining of Aldh1a1 confirmed the reductions in these two proteins in the central cornea (*Figure 7C, D*).

To examine whether a reduction in RA synthesis is involved in age-related corneal degeneration, we treated 20-month-old normal mice with RA for 4 months and examined their corneas. None of the RA-treated mice developed corneal plaques, and the percentage of mice with thinned corneas was greatly reduced (*Figure 7E*). Moreover, the alteration in histological structure and expression of epidermal marker genes in the aged cornea was also rescued (*Figure 7F* and *Figure 7—figure supplement 1*). These results confirm a critical role of RA in natural aging of the cornea and suggest the RA-containing ointment may be applied to the central cornea to slow down the development of aging-related corneal anomalies.

## Discussion

Corneal injuries are common due to the increasing use of contact lenses, laser-assisted in situ keratomileusis, air pollution, allergies, and infections (*Vashist et al., 2016*). Aberrant repair often leads to impaired vision, and under severe conditions, blindness becomes inevitable, and corneal replacement is needed (*Notara et al., 2010*). Thus, a complete understanding of the biology of corneal homeostasis and regeneration is imperative for developing regimens to preserve the cornea and treat cornea-related disorders. In this study, we established NF-κB, which responds to many extracellular stimuli and injury, as a critical regulator of corneal epithelial maintenance and regeneration. While the absence of *Rela* in K14$^+$ CESCs did not affect corneal development, it caused defective epithelial cell proliferation and differentiation and led to epithelial disruption at the central cornea in a cell-autonomous manner, which triggered inflammatory responses and led to epithelial metaplasia and plaque formation. This study thus uncovered the important mechanisms governing corneal homeostasis and regeneration.

The findings of our study suggest that centrally located CESCs play a role in corneal homeostasis and regeneration. We showed that activation of NF-κB was stronger in the central cornea than in the limbus. Deletion of *Rela* in K14$^+$ cells caused thinning and disruption of the corneal epithelium at the central cornea but not the limbus. Subsequently, epidermal fate conversion and plaque

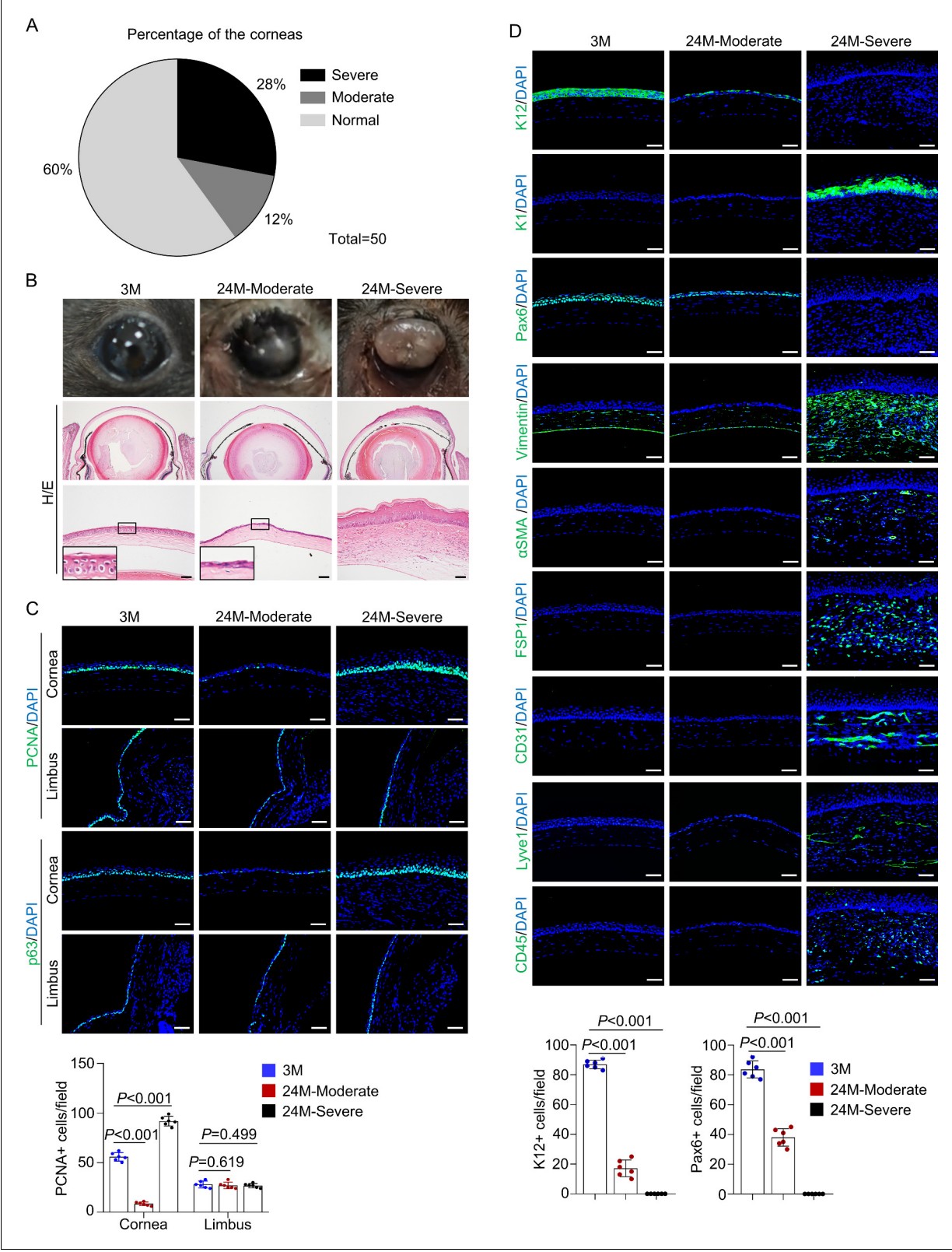

**Figure 6.** Natural aging produces similar corneal phenotypes as *Krt14-Cre; Rela*[f/f] mice. (**A**) The percentages of mice showing corneal epithelial layer thinning (moderate) and plaque formation (severe) at 24 months of age compared to young mice. n = 50. (**B**) Representative histological images showed thinning of corneal epithelial layer or overgrowth of corneal epithelial layer in old mice. Scale bar, 50 μm. (**C**) Representative immunostaining results showed alterations in the number of PCNA[+] cells and p63[+] cells while the limbus showed no alteration in 24-month-old mice (with cornea

*Figure 6 continued on next page*

*Figure 6 continued*

defects). Scale bar, 50 µm. Lower panel: quantitation data of PCNA$^+$ cells. n = 6 per group. (**D**) Representative immunostaining results showed alterations in the expression of differentiation marker K12, Pax6, epidermal marker K1, stromal cell marker vimentin, FSP1 and αSMA, blood vessel marker CD31, lymphatic vessel marker Lyve1, and immune cell marker CD45 in 24-month-old mice (with plaque formation). Scale bar, 50 µm. Lower panel: quantitation data. n = 6 per group. Data was presented as mean ± SEM. Unpaired two-tailed Student's t-test was applied in (**C, D**). p-value<0.05 was considered as statistically significant.

The online version of this article includes the following source data and figure supplement(s) for figure 6:

**Source data 1.** Numeric data used in *Figure 6*.
**Figure supplement 1.** Apoptosis rate was not affected in the cornea of 24-month-old mice.
**Figure supplement 1—source data 1.** Numeric data used in *Figure 6—figure supplement 1*.
**Figure supplement 2.** The peripheral and limbal regions appeared unaltered in aged mice.

formation occurred at the central cornea without affecting limbal CESCs. Furthermore, aging-related corneal phenotypes developed at the central cornea. If limbal CESCs were solely responsible for corneal regeneration, exhaustion of limbal CESCs should have occurred in *Krt14-Cre; Rela^f/f* mice. Overall, these findings indicate that central CESCs play an important role in corneal homeostasis and regeneration (*Majo et al., 2008*; *Chang et al., 2011*; *Nasser et al., 2018*; *West, 2015*; *Sun et al., 2010*). The differential responses to *Rela* deletion in the central and limbal CESCs are likely due to their distinct niches as the central cornea is constantly exposed to the external cues including UV lights, infectious particles, chemicals, and dirt and is under different mechanical strains, compared to limbal cells (*Nowell et al., 2016*). Moreover, we show that *Rela* ablation does not affect other organs/tissues marked by K14 in mice.

Mechanistically, we showed that NF-κB promoted corneal epithelial progenitor proliferation and differentiation by controlling RA synthesis. NF-κB directly regulated the expression of Aldh1a1, a rate-limiting enzyme in the RA synthesis pathway. Previous studies have shown that RA plays a role in eye development, likely via the Wnt-β-catenin pathway (*Smith et al., 2018*; *Niederreither and Dollé, 2008*; *Larange and Cheroutre, 2016*), which is involved in corneal stromal cells (*Kumar and Duester, 2010*; *Smith et al., 2018*; *Nowell et al., 2016*). Vitamin A deficiency in humans causes xerophthalmia, night blindness, corneal ulcers, and other symptoms, especially in children. Our study highlights the importance of RA, a vitamin A metabolite, in maintaining healthy corneas in adults and in preventing corneal aging. RA has been approved to treat skin disorders including psoriasis and acne as well as acute lymphatic leukemia (*Tang and Gudas, 2011*; *Ferreira et al., 2020*). In addition, ablation of *Notch1* in corneal epithelial cells also caused epithelial metaplasia and plaque formation (*Vauclair et al., 2007*). There are a few differences between these two mouse models. The *Notch1*-deficient mice show a loss of meibomian glands and much reduced expression of retinol transporting protein CRBP1 but no thinning or loss of the epithelial layer compared to *Krt14-Cre; Rela^f/f* mice. Despite these differences, both models underscore the importance of RA pathway in cornea maintenance and regeneration.

Notably, RA appears to play an insignificant role in subsequent wound healing events, including epithelial metaplasia and plaque formation. Instead, we showed that stroma-secreted VEGF and/or neovascularization play a causal role in these events. Our findings reinforce the importance of preventing neovascularization in maintaining healthy corneas and suggest that in patients already showing corneal epithelial metaplasia VEGFR inhibitors may help to slow the pathological progression.

Our findings indicate that NF-κB signaling plays an anti-aging role in mouse cornea. On the other hand, NF-κB activation can increase the expression of senescence-associated secretory phenotype cytokines, including IL-6, IL-1β, and TNF-α, which promote inflammation and aging in many tissues (*Taniguchi and Karin, 2018*). Thus, the activity of NF-κB must be fine-tuned in corneal epithelial cells. What causes the decline in *Rela* expression and activation in aged corneal epithelial cells? As a transcription factor regulating various cellular events, NF-κB is tightly controlled via multiple mechanisms including feedback regulatory circuits (*Taniguchi and Karin, 2018*). Constant activation of the pathway by pro-aging factors, for example, reactive oxygen species and chronic inflammation, may disrupt the regulatory circuits and lead to reduced expression and activation of RelA. This certainly warrants further investigation.

In human, although corneal epithelia thinning and keratinization have been observed in the elderly, corneal plaque formation and loss of vision rarely occur (*Yang et al., 2014*; *Blackburn et al.,*

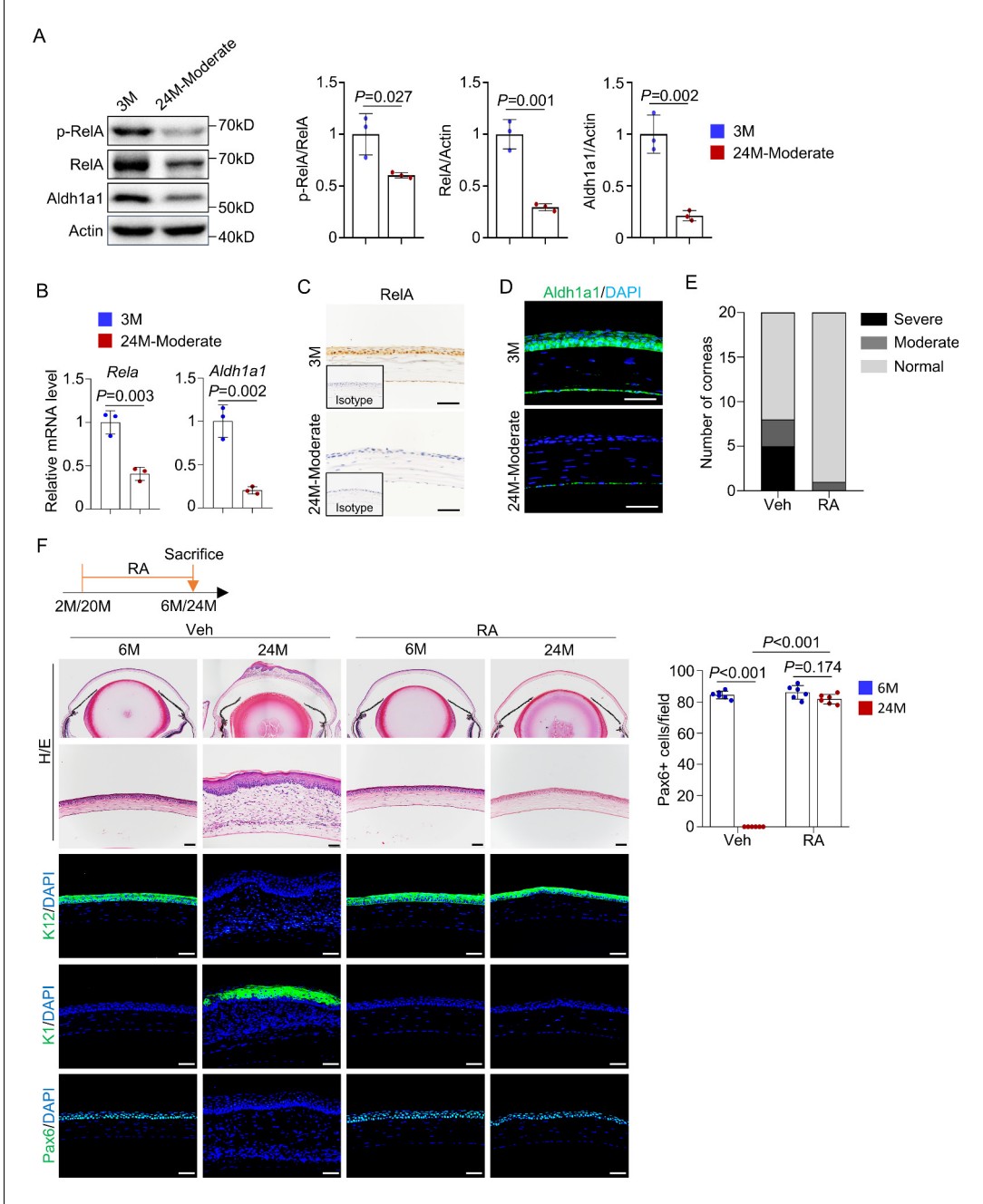

**Figure 7.** Retinoic acid (RA) supplementation prevents development of aging-like corneal phenotypes. (**A**) Representative western blot results showed decreased levels of RelA, p-RelA, and Aldh1a1 in the corneal samples of 24-month-old mice (with thinner epithelial layer) compared to young mice. Right panel: quantitation data. n = 3 per group. (**B**) Quantitative PCR analysis confirmed the decrease of *Rela* and *Aldh1a1* mRNA levels in the corneal epithelial samples of aged mice. n = 3 per group. (**C**) Representative immunohistochemical staining showed that the levels of RelA were decreased in the cornea of 24-month-old mice. Scale bar, 50 μm. (**D**) Representative immunostaining showed that levels of Aldh1a1 were drastically reduced in corneal samples of aged mice. Scale bar, 50 μm. (**E**) Administration of RA to 20-month-old normal mice alleviated the aging-related corneal phenotypes. n = 20 per group. (**F**) Representative histological results showed that RA prevented corneal defects in aged mice. Scale bar, 50 μm. Right panel: quantitation data. n = 6 per group. Data was presented as mean ± SEM. Unpaired two-tailed Student's t-test was applied in (**A**, **B**), and two-way ANOVA was applied in (**F**). p-value<0.05 was considered as statistically significant.

The online version of this article includes the following source data and figure supplement(s) for figure 7:

**Source data 1.** Numeric data used in *Figure 7*.
**Figure supplement 1.** Retinoic acid (RA) prevented increase of K14 in aged mice.

*2019*). One explanation is that aging-related decrease in RA synthesis may be made up for by RA-rich nutrients. Alternatively, age-related corneal problems may be diagnosed early and are treated successfully.

In summary, this study uncovers the important mechanisms governing corneal epithelium homeostasis, regeneration, and aging, and suggests that the canonical NF-κB-RA pathway may be a target to improve corneal health.

# Materials and methods

## Key resources table

| Reagent type (species) or resource | Designation | Source or reference | Identifiers | Additional information |
|---|---|---|---|---|
| Genetic reagent (*Mus musculus*) | Rela$^{f/f}$ | The Jackson Laboratory | RRID:IMSR_JAX:024342 | Stock No: 024342 |
| Genetic reagent (*M. musculus*) | Krt14-cre | The Jackson Laboratory | RRID:IMSR_JAX:004782 | Stock No: 004782 |
| Genetic reagent (*M. musculus*) | Prrx1-cre | The Jackson Laboratory | RRID:IMSR_JAX:005584 | Stock No: 005584 |
| Genetic reagent (*M. musculus*) | ROSA26$^{fs-tdTomato}$ | The Jackson Laboratory | RRID:IMSR_JAX:007914 | Stock No: 007914 |
| Antibody | Rabbit monoclonal to K12 | Abcam | Cat# ab185627, RRID:AB_2889825 | IF(1:200) |
| Antibody | Rabbit monoclonal to vimentin | Abcam | Cat# ab92547, RRID:AB_10562134 | IF(1:100) |
| Antibody | Rabbit polyclonal to αSMA | Abcam | Cat# ab5694, RRID:AB_2223021 | IF(1:100) |
| Antibody | Rat monoclonal to CD31 | Abcam | Cat# ab56299, RRID:AB_940884 | IF(1:100) |
| Antibody | Rabbit polyclonal to CD45 | Abcam | Cat# ab10558, RRID:AB_442810 | IF(1:100) |
| Antibody | Rabbit monoclonal to p63 | Abcam | Cat# ab124762, RRID:AB_10971840 | IF(1:100) |
| Antibody | Rabbit monoclonal to FSP1 | Abcam | Cat# ab197896, RRID:AB_2728774 | IF(1:100) |
| Antibody | Rabbit monoclonal to p-EGF receptor (Tyr1068) | Cell Signaling Technology | Cat# 3777, RRID:AB_2096270 | IF(1:100) WB(1:1000) |
| Antibody | Rabbit monoclonal to p-Stat3 (Tyr705) | Cell Signaling Technology | Cat# 9145, RRID:AB_2491009 | IF(1:100) WB(1:1000) |
| Antibody | Mouse monoclonal to Stat3 | Cell Signaling Technology | Cat# 9139, RRID:AB_331757 | WB(1:1000) |
| Antibody | Rabbit monoclonal to p-Erk1/2 (Thr202/Tyr204) | Cell Signaling Technology | Cat# 9101, RRID:AB_331646 | IF(1:100) WB(1:1000) |
| Antibody | Mouse monoclonal to Erk1/2 | Cell Signaling Technology | Cat# 9107, RRID:AB_10695739 | WB(1:1000) |
| Antibody | Rabbit monoclonal to RelA | Cell Signaling Technology | Cat# 8242, RRID:AB_10859369 | IHC(1:100) CHIP(1:100) |
| Antibody | Rabbit monoclonal to p-RelA (Ser536) | Cell Signaling Technology | Cat# 3033, RRID:AB_331284 | WB(1:1000) |
| Antibody | Rabbit monoclonal to RelA | Cell Signaling Technology | Cat# 4764, RRID:AB_823578 | WB(1:1000) |
| Antibody | Rabbit monoclonal to p-Akt (Ser473) | Cell Signaling Technology | Cat# 4060, RRID:AB_2315049 | WB(1:1000) |
| Antibody | Rabbit monoclonal to Akt | Cell Signaling Technology | Cat# 9272, RRID:AB_329827 | WB(1:1000) |

*Continued on next page*

*Continued*

| Reagent type (species) or resource | Designation | Source or reference | Identifiers | Additional information |
|---|---|---|---|---|
| Antibody | Rabbit monoclonal to p-IKKα/β (Ser176/180) | Cell Signaling Technology | Cat# 2697, RRID:AB_2079382 | WB(1:1000) |
| Antibody | Rabbit polyclonal to IKKα | Cell Signaling Technology | Cat# 2682, RRID:AB_331626 | WB(1:1000) |
| Antibody | Rabbit polyclonal to K1 | BioLegend | Cat# 905601, RRID:AB_2565051 | IF(1:200) |
| Antibody | Rabbit polyclonal to Pax6 | BioLegend | Cat# 901302, RRID:AB_2749901 | IF(1:200) |
| Antibody | Mouse monoclonal to LYVE-1 | Reliatech | Cat#103PA50AG, RRID:AB_2876870 | IF(1:100) |
| Antibody | Rabbit polyclonal to ALDH1A1 | Proteintech | Cat# 15910-1-AP, RRID:AB_2305276 | IF(1:100) |
| Antibody | Rabbit polyclonal to Ki-67 | Thermo Fisher Scientific | Cat# PA5-19462, RRID:AB_10981523 | IF(1:100) |
| Antibody | Mouse monoclonal to PCNA | Santa Cruz Biotechnology | Cat# sc-56, RRID:AB_628110 | IF(1:100) |
| Antibody | Rabbit polyclonal to β-catenin | Santa Cruz Biotechnology | Cat# sc-7199, RRID:AB_634603 | WB(1:1000) |
| Antibody | Mouse monoclonal to β-actin | Santa Cruz Biotechnology | Cat# sc-47778, RRID:AB_626632 | WB(1:5000) |
| Antibody | Anti-rabbit IgG, HRP-linked Antibody | Cell Signaling Technology | Cat# 7074, RRID:AB_2099233 | WB(1:5000) |
| Antibody | Anti-mouse IgG, HRP-linked Antibody | Cell Signaling Technology | Cat# 7076, RRID:AB_330924 | WB(1:5000) |
| Antibody | Goat anti-Rabbit IgG Secondary Antibody, Alexa Fluor488 | Thermo Fisher Scientific | Cat# A-11008, RRID:AB_143165 | IF(1:200) |
| Antibody | Goat anti-Mouse IgG Secondary Antibody, Alexa Fluor488 | Thermo Fisher Scientific | Cat# A-11001, RRID:AB_2534069 | IF(1:200) |
| Antibody | Goat anti-Rat IgG Secondary Antibody, Alexa Fluor488 | Thermo Fisher Scientific | Cat# A-11006, RRID:AB_2534074 | IF(1:200) |
| Sequence-based reagent | RT-qPCR primers | This paper | | See *Table 1* |
| Sequence-based reagent | CHIP primers | This paper | | See *Table 2* |
| Commercial assay or kit | PrimeScript RT reagent Kit | TAKARA | RR037A | |
| Commercial assay or kit | Fast Start Universal SYBR Green Master kit | Roche | 04887352001 | |
| Commercial assay or kit | SimpleChIP Enzymatic Chromatin IP Kit | Cell Signaling Technology | #9002 | |
| Chemical compound, drug | Retinoic acid | Sigma-Aldrich | Cat# R2625 | |
| Chemical compound, drug | BMS493 | Sigma-Aldrich | Cat# B6688 | |
| Chemical compound, drug | Axitinib | Selleck Chemicals | Cat# S1005 | |
| Chemical compound, drug | 0.5% Levofloxacin Eye Drops | Santen | N/A | |
| Chemical compound, drug | EZ-Link Sulfo-NHS-LC-Biotin | Thermo Fisher Scientific | Cat# A39257 | |
| Software, algorithm | ImageJ | (http://imagej.nih.gov/ij/) | | |
| Software, algorithm | GraphPad Prism 8 | https://www.graphpad.com | RRID:SCR_015807 | Version 8 |

## Mouse lines and maintenance

All the mouse work was carried out following the recommendations by the National Research Council Guide for the Care and Use of Laboratory Animals, with the protocols approved by the Institutional Animal Care and Use Committee of Shanghai, China [SYXK(SH)2011-0112]. *ROSA26^{fs-tdTomato}*, *Rela^{f/f}*, *Krt14-Cre*, *Prrx1-Cre* mouse lines were purchased from The Jackson Laboratory. Theses mice were crossed to B6 mice for at least five times. The young and aged normal mice were on B6 background and maintained at the same facility as the engineered mice.

## H/E staining, Alcian blue staining, and Picrosirius red staining

After the mice were euthanized, the whole eyeball was fixed in 4% paraformaldehyde overnight. Samples were then dehydrated, embedded in paraffin, sectioned at 4-μm-thick, and prepared for staining. H/E staining, Alcian blue staining and Picrosirius red staining were carried out following the standard protocols.

## Lineage tracing

All the lineage tracing experiments were done in both male and female adult mice with similar results obtained. To trace tdTomato-positive cells, cryostat sections were used. Mouse eyeballs were fixed and embedded in OCT (Leica) and frozen in liquid nitrogen. Sections were cut at 5 μm thick at −20℃, which were counterstained with DAPI solution and the images were taken under Microscope (Nikon ECLIPSE 80i).

## Immunofluorescent microscopy

The paraffin sections were dewaxed and rehydrated, then permeabilized with Triton X100 (0.1%) in PBS for 10 min. Antigen retrieval was performed with heated sodium citrate solution. The samples were blocked with 10% goat serum at room temperature (RT) for 1 hr, then incubated with primary antibodies at 4℃ overnight. For cryostat section, the frozen sections were rewarmed, washed, permeabilized with Triton X100 (0.1%) in PBS for 10 min, blocked with 10% goat serum at RT for 1 hr, and incubated with primary antibody at 4℃ overnight. The next day, after washing, the samples were incubated with secondary antibody at 37℃ for 1 hr, then counterstained with DAPI solution, and the images were taken under Microscope (Nikon ECLIPSE 80i). The antibodies are listed in Key resources table. Apoptosis was determined by TUNEL assay (In Situ Cell Death Detection Kit Fluorescein, Roche) following the standard protocols.

## Immunohistochemical staining

The paraffin sections were dewaxed and rehydrated. The samples were treated with 3% $H_2O_2$ for 10 min to quench the endogenous peroxidase activity, permeabilized with Triton X100 (0.1%) in PBS for 10 min. Antigen retrieval was performed with heated sodium citrate solution. The samples were blocked, then incubated with primary antibodies, secondary antibodies and third antibodies that conjugated to horseradish peroxidase (HRP). The samples were incubated with DAB solution (Boster), washed, and then counterstained with hematoxylin solution. The slides were then dehydrated, cleared, and mounted with resin for observation. See Key resources table for antibodies used. To quantify the RelA signals, cells with different DAB signal intensities were given the following scores: high (*Collinson et al., 2004*), positive (*Nowell and Radtke, 2017*), low positive (*Yazdanpanah et al., 2017*), and negative (0). We took the average score of all cells in the field as an indicator of RelA expression.

## Immunoblotting

Immunoblotting was performed with standard procedures. Proteins were extracted from the whole cornea or the epithelial layers using RIPA lysis buffer supplemented with protease inhibitor and phosphatase inhibitor. Samples were separated with 10% SDS-PAGE and transferred onto PVDF membranes (Millipore). Membranes were blocked with 5% milk for 1 hr and incubated with specific antibodies overnight at 4℃. The primary antibodies are listed in Key resources table. To quantify western blot data, we measured the density of each band by ImageJ, which was normalized to the loading control of the sample. The average of the experimental group samples was normalized to that of the normal or Veh-treated samples.

## Model of corneal alkaline burn

The mice were anesthetized with sodium pentobarbital (Sigma-Aldrich). A 1.0-mm-diameter circular piece of filter paper, soaked in 1 N NaOH, was placed on the central surface of the cornea for 10 s. After alkali exposure, the ocular surface was rinsed with sterile saline solution for 60 s. We used 3–6 mice per group based on similar studies in the literature.

## Corneal fragility assay

Experimental animals were anesthetized with intraperitoneal injections of sodium pentobarbital (Sigma-Aldrich). Under a stereomicroscope, a partial epithelial defect was created in both eyes by brushing with a wet Microsponge (Alcon) as described by Kao et al., 1996. Then eyeballs were removed and embedded in paraffin for histology. H/E-stained cornea sections were analyzed.

## Corneal tight junction integrity

Hutcheon et al., 2007 described a functional assay of corneal epithelial cell tight junction integrity using LC-biotin, which does not penetrate through the epithelium in the presence of intact tight junctions, whereas defective tight junctions allow penetration through the epithelium and into the corneal stroma. In brief, 10 µl LC-biotin staining solution (EZ-Link-Sulfo-NHS-LC-Biotin, 10 mM, Thermo Fisher Scientific) was applied to the cornea of wild-type and Rela-deficient mice for 15 min at the time of euthanasia. Eyeballs were rinsed with PBS, enucleated, and placed in OCT for frozen sectioning. Sections were stained with FITC-streptavidin to detect the presence of LC-biotin and then quantified based on FITC signals.

## RA and inhibitor administration

For in vitro experiments, RA (Sigma-Aldrich) and BMS493 (Sigma-Aldrich) were dissolved in dimethyl sulfoxide and applied at a final concentration of 10 mM and 5 mM in cell culture medium, respectively. For in vivo experiments, RA was dissolved in corn oil while axitinib (Selleck Chemicals) was dissolved in dimethyl sulfoxide, which were reconstituted in PEG300/Tween80/Water mixed solution. Mice were treated with 1 mg/kg RA or 10 mg/kg axitinib with the diluent as a control through intragastric administration every other day, following the experimental protocols. For antibiotic treatment, two drops of 0.5% Levofloxacin Eye Drops (Santen) were applied into each eye every day.

## Mouse primary corneal epithelial cell culture

Briefly, corneas were carefully dissected to ensure that conjunctival and iris tissues were not included and cut into two pieces and placed into a cell culture dish. After 10 min, the proliferation medium (Defined Keratinocyte-serum free medium [KSFM with growth factors], Life Technologies) was added into dish and cultured at 37°C in a 5% $CO_2$ incubator. Epithelial cells began to climb out from the tissue blocks. The proliferation medium was changed every 3 or 4 days. For differentiation, the cells were switched to differentiation medium (DMEM/F12 plus 10% FBS).

## Quantitative PCR analysis

Corneas were placed in 50 µl RNA stabilization reagent (RNAlater, Sigma-Aldrich), and later incubated in 250 µl 20 mM EDTA (sterile, pH 7.4) at 37°C for 30 min. The epithelial layer was then teased apart from the stromal layer. Total RNA was extracted from corneal epithelia, stroma or cultured cells samples with Trizol Reagent (Invitrogen) following the manufacturer's instruction. cDNA was reversed transcribed using PrimerScript RT reagent Kit (Takara), and quantitative PCR was carried out using the Roche Light Cycle 480II detection system (Roche). Primer sequences are shown in Table 1.

## ChIP assay

The ChIP assay was carried out following the manufacturer's protocol (SimpleChIP Enzymatic Chromatin IP Kit, Cell Signaling, 9002). Briefly, primary corneal epithelial cells were crosslinked with 1% formaldehyde and blocked with glycine, washed, and digested by micrococcal nuclease. The nuclear pellet was suspended in ChIP buffer and sheared by sonication. An aliquot of sheared chromatin sample was set aside as input control. The remained chromatin was then incubated with anti-RelA antibody (Cell Signaling, 8242). Normal Rabbit IgG (Cell Signaling, 2729) was used as a negative

**Table 1.** Quantitative PCR primer sequences used in the study.

| Gene | Forward primer | Reverse primer |
|---|---|---|
| Krt12 | CATGGCTGAGCAAAATCGGAA | CAGGGACGACTTCATGGCG |
| Rela | AGGCTTCTGGGCCTTATGTG | TGCTTCTCTCGCCAGGAATAC |
| Aldh1a1 | ATACTTGTCGGATTTAGGAGGCT | GGGCCTATCTTCCAAATGAACA |
| Krt1 | TGGGAGATTTTCAGGAGGAGG | GCCACACTCTTGGAGATGCTC |
| Krt10 | CGAAGAGCTGGCCTACCTAAA | GGGCAGCGTTCATTTCCAC |
| Vegfa | GCACATAGGAGAGATGAGCTTCC | CTCCGCTCTGAACAAGGCT |
| Vegfb | GCCAGACAGGGTTGCCATAC | GGAGTGGGATGGATGATGTCAG |
| Vegfc | GAGGTCAAGGCTTTTGAAGGC | CTGTCCTGGTATTGAGGGTGG |
| Vegfd | TTGAGCGATCATCCCGGTC | GCGTGAGTCCATACTGGCAAG |

control. The immunoprecipitated chromatins were then eluted with ChIP elution buffer. The DNA fragments were released by treatment with proteinase K at 65℃. The released DNA fragments were purified with columns and amplified by site-specific primers by quantitative PCR assay. The data were analyzed by the following formula: percent (%) input recovery = (100/(input fold dilution/bound fold dilution)) × 2(input CT - bound CT). Pairs of primers designed to amplify the specific target sequences of the putative promoters are listed in *Table 2*.

## RNA sequencing

To perform RNA sequencing, corneal epithelial layer was separated and total RNA was extracted from three independent biological samples. rRNA depletion was performed using Ribo-Zero Gold rRNA removal kit H/M/R (Illumina). Agilent RNA 6000 Nano Kit on 2100 Bioanalyzer (Agilent) was used to do total RNA sample QC. Libraries were constructed with a series of standard steps, included RNA fragment and reverse transcription to double-strand cDNA, end repair, tailing and adaptor ligation, PCR amplification, denaturation, and cyclization. The final library was single-strand circle DNA (ssCir DNA) and was amplified with phi29 (Thermo Fisher Scientific) to make DNA nano-ball (DNB). DNBs were transformed to single-end 50 bases reads and sequenced on BGISEQ-500 platform (BGI-Shenzhen, China).

## RNA-seq analysis

We used SOAPnuke (v1.5.2) to filter reads and generate FASTQ format (*Cock et al., 2010*). We aligned clean reads to the reference mouse genome using the Bowtie2 (v2.2.5) with default parameters and calculated gene expression level with *RSEM* (v1.2.12) (*Langmead and Salzberg, 2012*; *Li and Dewey, 2011*). Statistically significant genes (adjusted p-value≤0.001) with large expression changes (fold change ≥2) were defined as differentially expressed genes (DEGs) (*Wang et al., 2010*). In GO analysis and pathway analysis, DEGs were classified according to official classification with the GO or KEGG annotation results and phyper (a function of R) was performed in GO and pathway functional enrichment. The false discovery rate (FDR) was calculated for each p-value with FDR ≤ 0.01 defined as significantly enriched.

## Statistical analysis

Numerical data and histograms were expressed as the mean ± SEM. The number of mice used for each experiment was specified in the figure legends. For histology analysis, at least three mutant

**Table 2.** Primer sequences used for chromatin immunoprecipitation assays.

| Gene | Predictive binding site | Forward sequence | Reverse sequence |
|---|---|---|---|
| Aldh1a1 | S1: GCGAATTTCC | AACATCTTGGGGTGCATTGC | TAGCTAGGGGAGGAACAGGG |
| | S2: GGGACTTTTC | ATGATTCACAAGTGCACGCA | CAGAATCTTCGCATTGTCTTTGT |
| | S3: GGGATCTTCC | TGTTTGGGAATTGGCCTGAG | AGCCTGCTTCTCTCTCTCTC |

and control mice were used and representative results were presented and the number of mice used for each experiment was based on previously reported studies. Quantitative PCR was performed using RNA isolated from three mice. Western blot was performed using samples from three mice as well. Comparisons between two groups were analyzed using two-tailed unpaired Student's t-test, and p-value<0.05 was considered as statistically significant. All the experiments were repeated at least three times.

## Acknowledgements

The work was supported by the National Key Scientific Program (2018YFA0800801 and 2018YFA0800803) and the National Natural Science Foundation of China (81520108012 and 91749201).

## Additional information

### Funding

| Funder | Grant reference number | Author |
|---|---|---|
| National Key Research and Development Program of China | 2018YFA0800801 | Jing Li |
| National Key Research and Development Program of China | 2018YFA0800803 | Baojie Li |
| National Natural Science Foundation of China | 91749201 | Baojie Li |
| National Science Foundation of China | 81873679 | Jing Li |

The funders had no role in study design, data collection and interpretation, or the decision to submit the work for publication.

### Author contributions

Qian Yu, Conceptualization, Data curation, Formal analysis, Validation, Investigation, Visualization, Methodology, Project administration, Writing - review and editing; Soma Biswas, Methodology, Writing - review and editing; Gang Ma, Resources, Writing - review and editing; Peiquan Zhao, Writing - review and editing; Baojie Li, Conceptualization, Supervision, Funding acquisition, Writing - original draft, Project administration, Writing - review and editing; Jing Li, Conceptualization, Supervision, Project administration, Writing - review and editing

### Author ORCIDs

Soma Biswas http://orcid.org/0000-0002-1427-2678
Baojie Li https://orcid.org/0000-0002-3913-1062

### Ethics

Animal experimentation: All the mouse work was carried out following the recommendations by the National Research Council Guide for the Care and Use of Laboratory Animals, with the protocols approved by the Institutional Animal Care and Use Committee of Shanghai, China [SYXK(SH)2011-0112].

### Decision letter and Author response

Decision letter https://doi.org/10.7554/eLife.67315.sa1
Author response https://doi.org/10.7554/eLife.67315.sa2

## Additional files

### Supplementary files
• Transparent reporting form

### Data availability

Sequencing data have been deposited in GEO database (NCBI) under the accession Series GSE161433.

The following dataset was generated:

| Author(s) | Year | Dataset title | Dataset URL | Database and Identifier |
|-----------|------|---------------|-------------|-------------------------|
| Yu Q, Biswas S, Ma G, Zhao P, Li B, Li J | 2021 | Next Generation Sequencing Facilitates Quantitative Analysis of Wild Type and RELA-/- Cornea Transcriptomes | https://www.ncbi.nlm.nih.gov/geo/query/acc.cgi?acc=GSE161433 | NCBI Gene Expression Omnibus, GSE161433 |

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
