## [Decision Letter]

**Acceptance summary:**

This study thus uncovers major mechanisms governing maintenance of corneal transparency, regeneration and aging involving the retinoic acid pathway. The authors identify retinoic acid as a potential therapeutic for corneal disorders.

**Decision letter after peer review:**

Thank you for submitting your article "Canonical NF-κB signaling maintains corneal epithelial integrity and prevents corneal aging via retinoic acid" for consideration by *eLife*. Your article has been reviewed by 3 peer reviewers, and the evaluation has been overseen by a Reviewing Editor and Kathryn Cheah as the Senior Editor. The reviewers have opted to remain anonymous.

Essential revisions:

The manuscript by Yu and co-authors investigates the effects of ablating Rela (a subunit of NF-ƙB) in K14+ basal cells then studying how corneas heal after inflicting a chemical wound and the pathology seen after a substantial follow-up period. Some of the features that develop in the mutant mice seem to be recapitulated in a fraction of normal aged mice, implying that this factor is a key regulatory element in supporting corneal homeostasis. The authors provide a wealth of data using a wide array of techniques in relations to characterizing the pathways and pathway elements that partake in the interesting phenomenon. The major concerns are:

1. The lack of clear organization of the paper makes it difficult to follow and to understand the significance. Rewriting the introduction, study design, results and discussion as well as addressing missing or inappropriate controls, animal numbers, statistics and general clarifications would help to improve the quality and impact of this study.

2. The introduction should lead the reader into understanding why the study was done. In the discussion authors should not reiterate results but talk about 1. mechanism 2. significance with respect to human disease as age-related corneal dysfunction is not really a problem in humans without an underlying disease 3. how specific this discovery really is to the eye given that the mutation is global i.e. it presumably affects all 14+ basal epithelial cells through the body 4. The study indicates that there is a change in the level of Rela over time and that rescue can occur by administering retinoic acid. It does not explain why the enzyme decreases or is suppressed with age, nor does it describe why it is region specific.

3. The paper's organization may also be improved by moving some studies to supplemental data.

The studies on vegf and those on the treatment with antibiotics should be in the supplemental data as nothing more is done with them.

The studies on aged mice are interesting but in showing that the response is similar without doing RNA seq on both tissues and comparing them, the reader is left hanging. These could be moved to supplemental data.

4. Many references are missing.

5. Quantification of data is missing in some instances and some controls are missing as noted in the reviews. The specific comments in the reviews should help guide the authors.

*Reviewer # 1:*

This is a well executed study on the significance of knocking out Rela, that encodes a NFkB subunit in K14+ corneal epithelial stem cells in mice. This knockout results in aberrant wound healing, neovascularization, epithelial metaplasia, and plaque formation in the central cornea. Vitamin A deficiency is known to cause corneal dysfunction defects. Here the authors demonstrate that retinoic acid, a metabolic product of vitamin A, is important for regeneration and aging. The data provided by the authors largely support their conclusions. The main strength of the paper is novelty of demonstrating a RA-related defect in the central cornea but not the limbus. Furthermore, that normal aged corneas in mice have a similar defect to young Rela null mice, both of which are rescued with RA treatment.

In summary, the Rela KO did not affect corneal development. However, at 4 weeks when wounded with an alkaline burn, compared to control corneas, the Rela null mice demonstrate a loss of K12 and reduced PCNA in the cornea. Furthermore, after 2 months of aging (without injury), Rela null mice displayed large central plaque formation with neovascularization and metaplasia that increased in severity with aging. The limbus was not affected. These data suggest that Rela is required for maintaining corneal homeostasis mainly in the central cornea. Furthermore, RNA-seq analysis of the epithelial layers confirmed differentiation of epithelium into epidermis and cell cycle genes supporting excessive proliferation. Furthermore, using a K14-Cre; Rela^f/f^; Tomato mouse, the authors found that the epidermal genes were derived from K14+ cells. RA supplementation rescues the defect. Natural aging of mice demonstrates similar corneal defects to Rela null mice, which are rescued by treatment with RA.

The article could be strengthened in several ways. For example the interrogation of the importance of the VEGF pathway in plaque formation and metaplasia seems out of place. The authors treat for 6 months with an VEGFR inhibitor, although it does prevent neovascularization and metaplasia, it also prevents epithelial stratification. The analysis of these experiments would need to be clarified. Also, the VEGFR experiments are difficult to assimilate into the overall story and could be removed. In addition, the antibiotic treatment without an infection is difficult to comprehend and coordinate into the overall story. Finally, it would be helpful to propose how aging mouse corneas could compare to human corneal aging and if there is any data on corneal problems solely because of aging in humans.

Please mention that transparency in the cornea is also from the secretion of crystallins.

Please reference, limited by a lack of corneal donors, line 18.

The basement membrane is a critical contributor to the maintenance of epithelial/stromal biology. This should be incorporated into the introduction with references to papers by Steve Wilson, a leader in the basement membrane literature.

A fairer way to quantify western blot is to take the average of the control (which can then have standard error) and then find the values for experimental/ave control. Even though control will still be 1, it will have incorporated the error into control lanes. For the western in 1a, it would be Control/actin and then experimental/actin divided by ave of control/actin. By setting the control to 1 all error is being eliminated. A small graph might work better.

Mouse cornea is not fully developed until 6 weeks, the reason for NaOH at 4 weeks is not clear.

There are many studies on corneal alkaline injury and the role of NFkB, please better reference these studies.

It would have helpful to stain for scarring markers to understand more about the fibrotic response. There can be little SMA but still a large scar.

I'm not sure that one can conclude that (page 13 line 20) primary defects are caused by RA deletion and that the later effects are RA independent because these defects are not observed when the RA is supplied earlier.

Typo figure 7 "tatal".

Age-related corneal dysfunction is not really a problem in humans without an underlying disease. Please discuss.

*Reviewer #2:*

The authors used a number of approaches in studying the function of NFKB signaling in corneal epithelial homeostasis in control and K14-Cre;Rela^f/f^ mice. To generate these mice, Rela^f/f^ mice and K14-Cre mice were crossed and the following approaches were used to examine the mice over time and after injury; RNAseq, genetic analysis and lineage tracing, along with more traditional analyses of ChIP, molecular approaches and conventional immunohistochemistry. The authors showed that after an alkali injury there was a decrease in K12 and PCNA in the central cornea, that was not detected in the peripheral and limbal epithelium. Additional experiments supported the observation that the central cornea responds uniquely. When the authors examined the response in the stroma underlying the epithelium, they found that while there was an increase in thickness there was no fibrosis and they speculated that the response was secondary to the epithelium as it was not generated or detected until the epithelial changes occurred. RNA seq data of the cells where Rela was ablated demonstrated that there was a significant reduction in the expression of Aldh1a1 and ChIP verified the presence of the NFKB binding site on the Aldh1a1 promoter.

Further analyses demonstrated that the abnormal cells were all tomato+ indicating that the cells in the center were derived from K14; however it is not understood why the other K14 labelled cells in the cornea or in basal skin were not affected by RelA deletion. The authors showed that the response could be rescued when corneas were treated with retinoic acid before thinning and plaque development was detected. Conversely, there was no rescue if treated after thinning occurred. The authors also performed confirmatory experiments in vitro and found that cultured epithelial cells that were Rela^-/-^ displayed decreased proliferation and K12 and these could also be rescued with retinoic acid. These all indicate that NFKB signaling appears to regulate epithelial homeostasis and that the biological changes are presumably caused by a decrease in the enzyme, Aldh1a1, and can be rescued with exogenous retinoic acid. In addition these indicate that the environment of the central corneal cells is unique and the authors suggest that the stroma plays a major role but their investigation into this question are relatively limited and a mixture of questions are asked. That being said, the study up to this point is extremely interesting and suggests that the pathology is dynamic and "flexible" up until a certain point. While this does demonstrate that the study is not complete, it also suggests that there are many facets to further investigate, most specifically the role of the stroma.

As mentioned above, the weakness of the study occurred when the authors performed a number of experiments to understand the source of the factor. These included an analysis of expression of VEGF isoforms along with an inhibitor of VEGFR and an antibiotic. Neither of these altered the specific response and so only provided more questions. That being said, a simple ageing study revealed a very interesting observation and that is there was a natural thinning of the corneal thinning by 24 months of age that was similar to that induced by ablation of RelA in the mice. The authors showed that the changes were not detected at 20 months and demonstrated that if mice were treated with retinoic acid at 20 months for four months, they could delayed or inhibit thinning or plaque development.

The study is quite interesting where a normal ageing study is compared to changes that occur when RelA is ablated. The study indicates that there is a change in the level of enzyme over time and that rescue can occur by administering retinoic acid. It does not explain why the enzyme decreases or is suppressed with age, nor does it describe why it is region specific. The manuscript would be improved by:

1. Higher magnification of central and peripheral corneal images of the exact time pts where thinning occurs for the different conditions: Rela ablation, alkali burn, ageing in the presence or absence of RA. Digital expansion revealed a certain level of resolution but was not sufficient.

2. Supplement figures 1 and 2 should go in text to show the regions of the cornea that are affected. They confirm and are important.

3. Add the data in supplement figure 3 of corneal defects to figure 2 as it rules out a number of questions that occurs while the manuscript is being read and it completes the data.

4. The experiments on Vegf and antibiotics might fit better in supplement as they did not describe the mechanism.

5. The summary sentence of the result is weak and similarly for the discussion. It is quite an interesting and complex study and additional explanations should be explored! It appears that the canonical NFKb signaling pathway is a potential target, but it is complex as expression of K12 is not the only predictor as cells that are affected are in the central cornea and not peripheral or limbal.

*Reviewer #3:*

The manuscript by Yu and co-authors details an investigation into the effects of ablating Rela (a subunit of NF-ƙB) in K14+ basal cells then studying how corneas heal after inflicting a chemical wound and the pathology that ensure after a substantial follow-up period. Some of the features that develop in the mutant mice seem to be recapitulated is a fraction of physicologically aged (normal) mice, implying that this factor is a key regulatory element in supporting corneal homeostasis. The authors provide a wealth of data using a wide array of techniques in relations to characterizing the pathways and pathway elements that partake in the interesting phenomenon. This reviewer wondered how specific to the eye this discovery really is, given the mutation is global i.e. it presumably affects all 14+ basal epithelial cells through the body.

I have several concerns related to the writing of several section, study design, omission or inappropriate controls, animal numbers, statistics and general clarifications that should be addressed to improve the quality and impact of this study.

1. Introduction, page 4, paragraph 2, lines 7-8: "We hypothesized that NF-ƙB signaling plays important physiological roles in the cornea" Are there prior notifications in the literature that support this proposition? If so, the authors should elaborate and cite the relevant literature.

2. Introduction, page 4, paragraph 2, lines 8-22: This entire commentary is devoted to rehashing the results of the study. Overall, I felt the Introduction was poorly written because it provided little or no direction as to where the research was heading.

3. Results, page 6, lines 3-4: This first statement is an example of what is needed in the Introduction. This is to inform the reader of what is to come in the manuscript.

4. Results, related to Figure 1, lines 7 and 9: "but not much" and "reduced extent" We do not know for certain whether these results are significant or not because they are descriptive. I suggest some form of quantitation be performed.

5. Results, related to Figure 1, line 10: In reference to 'stromal' cells; what is their phenotype, i.e. are they resident keratocytes or infiltrating inflammatory cells?

6. Results, related to Figure 1, page 7, lines 1 and 2: "lacked a basal layer and basement membrane" How do the authors reconcile the absence of a BM without having stained for BM components or indeed performing ultrastructural (i.e. EM) investigations?

7. Figure 1: Please provide the graphical representation for the Western Blot in panel a and the appropriate statistics. In relation to Figure 1 panel b please provide the appropriate data from the isotype controls, this will establish specificity of the Rel A staining. IN relation to Figure 1 panel c. Please provide the quantification and the number of independent replicate this blot represents.

8. Results, related to Figure 2, line 8: 'Prrx1-cre', perhaps I missed it in my read but was this line ever explained?

9. Results, related to Figure 2, page 8, paragraph 1: In relation to the K14-Cre Rela^f/f^ line are there any aberrations in other organs apart from the eye/cornea after all there are K14+ basal cells that are present in other organs, skin, nasal, oral mucosa, conjunctiva, intestine, etc. I realise that they have investigated the skin of these mice and found not abnormalities (Suppl Figure S5) but how old were these mice compared to those which develop corneal defects?

10. Results, related to Figure 3, page 8, line 20: "we found stromal cell hyperproliferation, leukocyte infiltration,.…." How were stromal cells marked, and were they activated fibroblasts i.e. myofibroblasts?

11. Results, page 9, line 12: "which are features characteristic of hyperproliferative skin" One way of confirming the phenotypic switch is to stain for pax6.

12. Results, related to Figure 3, page 10, paragraph 1: The authors should stain for K14 in wt and their mutant mice to confirm that the expression of this marker is not altered due to the genetic alteration in mutant mice.

13. Results, related to Figure 4, page 12, paragraph 1, line 8: Is RA synthesis defective in any other ocular tissue of body organ or (again) is this a cornea specific anomaly? The authors should screen other sites.

14. Results: In relation to Figure 5, none of the observations have been quantified, an effect is assumed based on visual inspection.

15. Results, related to Figure 6b, page 14, paragraph 1, lines 9-12: Axitinib had multiple affects e.g it 'suppressed epithelial cell metaplasia' In this setting it may have suppressed K1 but did K12 expression did not return. Therefore, what are these epithelial cells lining the cornea? Please identify and please quantify all features displayed in Figure 6b.

16. Results, related to Figure 7, page 16, paragraph 1: In relation to the data presented in Figure 7 are the naturally aged normal mice non-genetically manipulated (i.e. truly wt) and are they bread on the same background to the mutant mice? If not, this is a major flaw in the study. Did any BV or lymphatics form in the physiologically aged mice that developed corneal lesions and what is the pax6 status of these corneas? Likewise, some quantification of the data presented in panel d is necessary. In the third column of panel d what is vimentin detecting versus the α-SMA?

17. Results, related to Figure 7: In general, if mutant mice were aged (without wounding) would they develop the same corneal lesions (and in the same proportion of individuals) as their physiologically aged counterparts?

18. Results, related to Figure 8: In relation to opacity, how was this feature defined and measured. I suggest the authors identify the phenotype of the epithelium covering the cornea after RA treatment (K12/pax6/K14/ etc)

19. Discussion, general comment: How different is the corneal pathology in Notch mutant mice (see reference 21) compared to the defects that develop in the line being described herein?

20. Discussion, page 20, paragraph 2, lines 5 and 13: Does the expression of epidermal-specific genes revert to corneal-like genes after RA application? In other words, once cells have committed switch phenotype, are they able to revert back to their original lineage? Likewise, addition of VEGFR inhibitors seems to restore some of the pathological features but can this treatment restore corneal epithelial phenotype? This information is important.

21. Materials and methods, IF microscopy, IHC staining, Immunoblotting, pages 22-24: Information related to antibodies usage should be tabulated, informing the reader of the dilution and or final concentration of primary and secondary Abs used in each assay.

22. Materials and methods, alkaline burn model, page 24: How many mice in total were used and how was this number (per group) arrived at?

23. Materials and methods, corneal tight junction integrity, page 25: Why was this readout not quantifiable?

24. Materials and methods, RA and VEGF inhibitor administration, page 25: What were the diluent controls. Were they DMSO and corn oil respectively?

---

## [Author Response]

Essential revisions:The manuscript by Yu and co-authors investigates the effects of ablating Rela (a subunit of NF-ƙB) in K14+ basal cells then studying how corneas heal after inflicting a chemical wound and the pathology seen after a substantial follow-up period. Some of the features that develop in the mutant mice seem to be recapitulated in a fraction of normal aged mice, implying that this factor is a key regulatory element in supporting corneal homeostasis. The authors provide a wealth of data using a wide array of techniques in relations to characterizing the pathways and pathway elements that partake in the interesting phenomenon. The major concerns are:1. The lack of clear organization of the paper makes it difficult to follow and to understand the significance. Rewriting the introduction, study design, results and discussion as well as addressing missing or inappropriate controls, animal numbers, statistics and general clarifications would help to improve the quality and impact of this study.

We have addressed these concerns accordingly in the revised manuscript. We have rewritten the introduction to highlight the study design, and the discussion. We have moved some of the results into the supplementary data, added the required controls, the numbers of mice used, and the statistics analysis results, and made some required clarifications.

2. The introduction should lead the reader into understanding why the study was done. In the discussion authors should not reiterate results but talk about 1.mechanism 2.significance with respect to human disease as age-related corneal dysfunction is not really a problem in humans without an underlying disease 3. how specific this discovery really is to the eye given that the mutation is global i.e. it presumably affects all 14+ basal epithelial cells through the body 4. The study indicates that there is a change in the level of Rela over time and that rescue can occur by administering retinoic acid. It does not explain why the enzyme decreases or is suppressed with age, nor does it describe why it is region specific.

We have shortened the description of the results and expanded the 4 aspects mentioned by the editor, in the discussion.

3. The paper's organization may also be improved by moving some studies to supplemental data.The studies on vegf and those on the treatment with antibiotics should be in the supplemental data as nothing more is done with them.The studies on aged mice are interesting but in showing that the response is similar without doing RNA seq on both tissues and comparing them, the reader is left hanging. These could be moved to supplemental data.

We have moved the results regarding VEGF and antibiotics into the supplementary information.

Regarding the aging-related studies, we think that it is an important part of the manuscript and hope to leave them in the main figures.

4. Many references are missing.

We have added 12 more references.

5. Quantification of data is missing in some instances and some controls are missing as noted in the reviews. The specific comments in the reviews should help guide the authors.

We have added the quantitation data wherever possible throughout the main and supplemental figures. Note that some of the staining results, for examples, extracellular proteins, are difficult to quantify.

Reviewer # 1:This is a well executed study on the significance of knocking out Rela, that encodes a NFkB subunit in K14+ corneal epithelial stem cells in mice. This knockout results in aberrant wound healing, neovascularization, epithelial metaplasia, and plaque formation in the central cornea. Vitamin A deficiency is known to cause corneal dysfunction defects. Here the authors demonstrate that retinoic acid, a metabolic product of vitamin A, is important for regeneration and aging. The data provided by the authors largely support their conclusions. The main strength of the paper is novelty of demonstrating a RA-related defect in the central cornea but not the limbus. Furthermore, that normal aged corneas in mice have a similar defect to young Rela null mice, both of which are rescued with RA treatment.In summary, the Rela KO did not affect corneal development. However, at 4 weeks when wounded with an alkaline burn, compared to control corneas, the Rela null mice demonstrate a loss of K12 and reduced PCNA in the cornea. Furthermore, after 2 months of aging (without injury), Rela null mice displayed large central plaque formation with neovascularization and metaplasia that increased in severity with aging. The limbus was not affected. These data suggest that Rela is required for maintaining corneal homeostasis mainly in the central cornea. Furthermore, RNA-seq analysis of the epithelial layers confirmed differentiation of epithelium into epidermis and cell cycle genes supporting excessive proliferation. Furthermore, using a K14-Cre; Rela^f/f^; Tomato mouse, the authors found that the epidermal genes were derived from K14+ cells. RA supplementation rescues the defect. Natural aging of mice demonstrates similar corneal defects to Rela null mice, which are rescued by treatment with RA.The article could be strengthened in several ways. For example the interrogation of the importance of the VEGF pathway in plaque formation and metaplasia seems out of place. The authors treat for 6 months with an VEGFR inhibitor, although it does prevent neovascularization and metaplasia, it also prevents epithelial stratification. The analysis of these experiments would need to be clarified. Also, the VEGFR experiments are difficult to assimilate into the overall story and could be removed. In addition, the antibiotic treatment without an infection is difficult to comprehend and coordinate into the overall story. Finally, it would be helpful to propose how aging mouse corneas could compare to human corneal aging and if there is any data on corneal problems solely because of aging in humans.

We appreciate the reviewer’s comments and suggestions. We have moved the results of VEGFR experiments and antibiotic experiments to the supplementary data.

We have clarified why epithelial stratification was rescued. We found that stroma of the mutant mice expressed much higher levels of VEGFA, which could stimulate proliferation of epithelial or epidermal cells, in addition to promoting angiogenesis. This phenomenon has been observed in liver/liver cancer cells and other tissues.

The reason why we did antibiotics experiments is that the cornea of the mutant mice have open wounds due to disruption of the epithelial layer, this will increase the risk of infection. Indeed, we found immune cell infiltration in the cornea, which was suppressed by antibiotics treatment.

We also compared cornea aging in mice and human in the discussion. There are a couple of studies showing corneal thinning and keratinization in aged individuals, although they do not show progression into plaque formation or blindness. One possible explanation is that retinoic acid deficiency is uncommon due to balanced diet in human. Another possibility is that aging-induced eye defects may be diagnosed early and are successfully treated.

Please mention that transparency in the cornea is also from the secretion of crystallins.

We appreciate the reviewer’s suggestion and added the information in the introduction.

Please reference, limited by a lack of corneal donors, line 18.

We have added the references.

The basement membrane is a critical contributor to the maintenance of epithelial/stromal biology. This should be incorporated into the introduction with references to papers by Steve Wilson, a leader in the basement membrane literature.

We appreciate the reviewer’s suggestion and added the information on basement membranes and cited Steve Wilson’ papers.

A fairer way to quantify western blot is to take the average of the control (which can then have standard error) and then find the values for experimental/ave control. Even though control will still be 1, it will have incorporated the error into control lanes. For the western in 1a, it would be Control/actin and then experimental/actin divided by ave of control/actin. By setting the control to 1 all error is being eliminated. A small graph might work better.

We appreciate the reviewer’s suggestions. We have quantified all the western blot results in the way suggested by the reviewer and presented the data in the revised manuscript (Figure 1A, Figure 1C, Figure 4E and Figure 7A).

Mouse cornea is not fully developed until 6 weeks, the reason for NaOH at 4 weeks is not clear.

We appreciate the reviewer’s concern. The *Rela* knockout mice developed corneal defects at 6 weeks of age. We wanted to test the effect of *Rela* ablation on corneal regeneration before they show any defects. That was why we did the NaOH treatment at the age of 4 weeks and analyzed the mice at 6 weeks of age. We have clarified this issue in the revised manuscript.

There are many studies on corneal alkaline injury and the role of NFkB, please better reference these studies.

We appreciate the reviewer’s suggestion and added more related references.

It would have helpful to stain for scarring markers to understand more about the fibrotic response. There can be little SMA but still a large scar.

We appreciate the reviewer’s suggestion. We have performed immunostaining of FSP1 and picrosirius red staining to show the fibrotic response. The new results were included in the revised manuscript (Figure 3A, Figure 3—figure supplement 1, and Figure 6D).

I'm not sure that one can conclude that (page 13 line20) primary defects are caused by RA deletion and that the later effects are RA independent because these defects are not observed when the RA is supplied earlier.

We appreciate the reviewer’s comments. What we meant was that metaplasia and plaque formation were not directly caused by RA deficiency. We have rewritten this part of text to clarify this.

Typo figure 7 "tatal".

We have corrected this error.

Age-related corneal dysfunction is not really a problem in humans without an underlying disease. Please discuss.

We appreciate the reviewer’s suggestion and have expanded this part of the discussion.

Reviewer #2:The authors used a number of approaches in studying the function of NFKB signaling in corneal epithelial homeostasis in control and K14-Cre;Rela^f/f^ mice. To generate these mice, Rela^f/f^ mice and K14-Cre mice were crossed and the following approaches were used to examine the mice over time and after injury; RNAseq, genetic analysis and lineage tracing, along with more traditional analyses of ChIP, molecular approaches and conventional immunohistochemistry. The authors showed that after an alkali injury there was a decrease in K12 and PCNA in the central cornea, that was not detected in the peripheral and limbal epithelium. Additional experiments supported the observation that the central cornea responds uniquely. When the authors examined the response in the stroma underlying the epithelium, they found that while there was an increase in thickness there was no fibrosis and they speculated that the response was secondary to the epithelium as it was not generated or detected until the epithelial changes occurred. RNA seq data of the cells where Rela was ablated demonstrated that there was a significant reduction in the expression of Aldh1a1 and ChIP verified the presence of the NFKB binding site on the Aldh1a1 promoter.Further analyses demonstrated that the abnormal cells were all tomato+ indicating that the cells in the center were derived from K14; however it is not understood why the other K14 labelled cells in the cornea or in basal skin were not affected by RelA deletion. The authors showed that the response could be rescued when corneas were treated with retinoic acid before thinning and plaque development was detected. Conversely, there was no rescue if treated after thinning occurred. The authors also performed confirmatory experiments in vitro and found that cultured epithelial cells that were Rela^-/-^ displayed decreased proliferation and K12 and these could also be rescued with retinoic acid. These all indicate that NFKB signaling appears to regulate epithelial homeostasis and that the biological changes are presumably caused by a decrease in the enzyme, Aldh1a1, and can be rescued with exogenous retinoic acid. In addition these indicate that the environment of the central corneal cells is unique and the authors suggest that the stroma plays a major role but their investigation into this question are relatively limited and a mixture of questions are asked. That being said, the study up to this point is extremely interesting and suggests that the pathology is dynamic and "flexible" up until a certain point. While this does demonstrate that the study is not complete, it also suggests that there are many facets to further investigate, most specifically the role of the stroma.As mentioned above, the weakness of the study occurred when the authors performed a number of experiments to understand the source of the factor. These included an analysis of expression of VEGF isoforms along with an inhibitor of VEGFR and an antibiotic. Neither of these altered the specific response and so only provided more questions. That being said, a simple ageing study revealed a very interesting observation and that is there was a natural thinning of the corneal thinning by 24 months of age that was similar to that induced by ablation of RelA in the mice. The authors showed that the changes were not detected at 20 months and demonstrated that if mice were treated with retinoic acid at 20 months for four months, they could delayed or inhibit thinning or plaque development.

We thought that the findings that VEGFR inhibitor could suppress corneal epithelial metaplasia and plaque formation may have some clinical implications. The reason why we did the antibiotics-related study is that we saw open wounds in the cornea of the mutant mice, which would increase the risk of infection. Indeed we see immune cell infiltration in the stroma of the mutant mice was suppressed by antibiotics treatment.

Nevertheless, we have moved these results into the supplementary data.

The study is quite interesting where a normal ageing study is compared to changes that occur when RelA is ablated. The study indicates that there is a change in the level of enzyme over time and that rescue can occur by administering retinoic acid. It does not explain why the enzyme decreases or is suppressed with age, nor does it describe why it is region specific. The manuscript would be improved by:

We have discussed the possibility why Rela expression is suppressed in the aged corneal epithelia. This is an important question and is definitely worth studying in the future.

1. Higher magnification of central and peripheral corneal images of the exact time pts where thinning occurs for the different conditions: Rela ablation, alkali burn, ageing in the presence or absence of RA. Digital expansion revealed a certain level of resolution but was not sufficient.

We appreciate the reviewer’s suggestions. We have mended the figures and uploaded high magnification images in the figures mentioned (Figure 1D, Figure 1—figure supplement 2D, Figure 2B, Figure 2—figure supplement 2, Figure 5E, Figure 5—figure supplement 1B, Figure 6B, Figure 6—figure supplement 2).

2. Supplement figures 1 and 2 should go in text to show the regions of the cornea that are affected. They confirm and are important.

We have fixed the figures as suggested.

3. Add the data in supplement figure 3 of corneal defects to figure 2 as it rules out a number of questions that occurs while the manuscript is being read and it completes the data.

We appreciate the reviewer’s suggestion and have moved these figures into the main Figure 2 in the revised manuscript.

4. The experiments on Vegf and antibiotics might fit better in supplement as they did not describe the mechanism.

We appreciate the reviewer’s recommendations. We have moved the results on Vegf and antibiotics to the supplementary data.

5. The summary sentence of the result is weak and similarly for the discussion. It is quite an interesting and complex study and additional explanations should be explored! It appears that the canonical NFKb signaling pathway is a potential target, but it is complex as expression of K12 is not the only predictor as cells that are affected are in the central cornea and not peripheral or limbal.

We appreciate the reviewer’s suggestions. We have rewritten the result summary sentences and the discussion. In addition, we have toned down the notion that RA is a target for treatment of related ocular diseases and proposed the use of RA-containing ointment to the central cornea.

Reviewer #3:The manuscript by Yu and co-authors details an investigation into the effects of ablating Rela (a subunit of NF-ƙB) in K14+ basal cells then studying how corneas heal after inflicting a chemical wound and the pathology that ensure after a substantial follow-up period. Some of the features that develop in the mutant mice seem to be recapitulated is a fraction of physicologically aged (normal) mice, implying that this factor is a key regulatory element in supporting corneal homeostasis. The authors provide a wealth of data using a wide array of techniques in relations to characterizing the pathways and pathway elements that partake in the interesting phenomenon. This reviewer wondered how specific to the eye this discovery really is, given the mutation is global i.e. it presumably affects all 14+ basal epithelial cells through the body.

We have analyzed a few tissues where epithelial cells are known to be marked by K14 and found that these tissues did not show obvious defects in the knockout mice, indicating that the effect of *Rela* ablation is rather specific to the central cornea. This may be attributable to constant exposure of the cornea, especially the central region, to external environment including UV lights, infectious particles, chemicals, and dirt.

I have several concerns related to the writing of several section, study design, omission or inappropriate controls, animal numbers, statistics and general clarifications that should be addressed to improve the quality and impact of this study.1. Introduction, page 4, paragraph 2, lines 7-8: "We hypothesized that NF-ƙB signaling plays important physiological roles in the cornea" Are there prior notifications in the literature that support this proposition? If so, the authors should elaborate and cite the relevant literature.

We appreciate the reviewer’s comments. We have rewritten this part of the introduction and added more references.

2. Introduction, page 4, paragraph 2, lines 8-22: This entire commentary is devoted to rehashing the results of the study. Overall, I felt the Introduction was poorly written because it provided little or no direction as to where the research was heading.

We appreciate the reviewer’s concern. We have rewritten this part of the introduction by shortening the description of the results and stressing the logic behind the study design.

3. Results, page 6, lines 3-4: This first statement is an example of what is needed in the Introduction. This is to inform the reader of what is to come in the manuscript.

We appreciate the reviewer’s concern. We have rewritten this part of the introduction and cited more papers.

4. Results, related to Figure 1, lines 7 and 9: "but not much" and "reduced extent" We do not know for certain whether these results are significant or not because they are descriptive. I suggest some form of quantitation be performed.

We appreciate the reviewer’s suggestions. We have added the quantitation data wherever possible.

5. Results, related to Figure 1, line 10: In reference to 'stromal' cells; what is their phenotype, i.e. are they resident keratocytes or infiltrating inflammatory cells?

They are keratocytes. We have clarified this in the revised text.

6. Results, related to Figure 1, page 7, lines 1 and 2: "lacked a basal layer and basement membrane" How do the authors reconcile the absence of a BM without having stained for BM components or indeed performing ultrastructural (i.e. EM) investigations?

We appreciate the reviewer’s concern. We have performed immunostaining of laminin (a marker of basement membrane) and the results support the disruption of the basement membranes (Figure 1D).

7. Figure 1: Please provide the graphical representation for the Western Blot in panel a and the appropriate statistics. In relation to Figure 1 panel b please provide the appropriate data from the isotype controls, this will establish specificity of the Rel A staining. IN relation to Figure 1 panel c. Please provide the quantification and the number of independent replicate this blot represents.

We appreciate the reviewer’s suggestions. We have quantified all western blot results and added the statistics data. We also added isotype control for RelA antibody and the information about the number of mice used in the experiments (Figure 1A-C).

8. Results, related to Figure 2, line 8: 'Prrx1-cre', perhaps I missed it in my read but was this line ever explained?

We have added information about Prrx1 and Prrx1-Cre mice.

9. Results, related to Figure 2, page 8, paragraph 1: In relation to the K14-Cre Rela^f/f^ line are there any aberrations in other organs apart from the eye/cornea after all there are K14+ basal cells that are present in other organs, skin, nasal, oral mucosa, conjunctiva, intestine, etc. I realise that they have investigated the skin of these mice and found not abnormalities (Suppl Figure S5) but how old were these mice compared to those which develop corneal defects?

In our initial submissions, we have analyzed the conjunctiva and meibomian glands in 6-month-old mice. In the revised manuscript, we have added the results for oral mucosa, intestine and trachea (see tracing results in Figure 1—figure supplement 2B and histological studies of *Rela* knockout mice in Figure 3—figure supplement 4C). There are no anomalies in these organs in 6-month-old mutant mice, confirming the effect of *Rela* ablation is rather specific to the central cornea. We have discussed the possibility for this tissue-specific effects in the revised manuscript.

10. Results, related to Figure 3, page 8, line 20: "we found stromal cell hyperproliferation, leukocyte infiltration,.…." How were stromal cells marked, and were they activated fibroblasts i.e. myofibroblasts?

We used Vimentin and FSP1 (new data) to mark stromal cells and αSMA to mark activated fibroblasts including myofibroblasts (Figure 3A).

11. Results, page 9, line 12: "which are features characteristic of hyperproliferative skin" One way of confirming the phenotypic switch is to stain for pax6.

We appreciate the reviewer’s suggestion. We have done immunostaining of Pax6 and found that Pax6 expression is drastically reduced in the corneas of the mutant mice (Figure 3F).

12. Results, related to Figure 3, page 10, paragraph 1: The authors should stain for K14 in wt and their mutant mice to confirm that the expression of this marker is not altered due to the genetic alteration in mutant mice.

We appreciate the reviewer’s suggestion. We have performed immunostaining of K14 on eye sections and found that K14 expression is normal in young mice before they develop corneal phenotypes (Figure 1—figure supplement 2C), suggesting that K14 is not affected by Cre insertion or *Rela* deletion. We also stained cultured corneal epithelial cells from WT and the mutant mice and found that K14 expression is unaltered (data not shown due to space limit).

13. Results, related to Figure 4, page 12, paragraph 1, line 8: Is RA synthesis defective in any other ocular tissue of body organ or (again) is this a cornea specific anomaly? The authors should screen other sites.

We appreciate the reviewer’s comments. We have done immunostaining of Aldh1a1 and found that expression of this enzyme was reduced in the limbus and conjunctiva of the mutant mice, yet, meibomian glands express undetectable levels of this enzyme in WT or mutant mice (Figure 4F, and Figure 4—figure supplement 2).

14. Results: In relation to Figure 5, none of the observations have been quantified, an effect is assumed based on visual inspection.

We appreciate the reviewer’s suggestions. We have added the quantification data (Figure 5A, 5D, and 5F)

15. Results, related to Figure 6b, page 14, paragraph 1, lines 9-12: Axitinib had multiple affects e.g it 'suppressed epithelial cell metaplasia' In this setting it may have suppressed K1 but did K12 expression did not return. Therefore, what are these epithelial cells lining the cornea? Please identify and please quantify all features displayed in Figure 6b.

We appreciate the reviewer’s concerns. There are no epithelial cells lining the cornea after axitinib treatment as epithelial layer disruption was not rescued by Axitinib. We have added quantification data and clarified this issue in the revised manuscript (Figure 5—figure supplement 2B).

16. Results, related to Figure 7, page 16, paragraph 1: In relation to the data presented in Figure 7 are the naturally aged normal mice non-genetically manipulated (i.e. truly wt) and are they bread on the same background to the mutant mice? If not, this is a major flaw in the study. Did any BV or lymphatics form in the physiologically aged mice that developed corneal lesions and what is the pax6 status of these corneas? Likewise, some quantification of the data presented in panel d is necessary. In the third column of panel d what is vimentin detecting versus the α-SMA?

We appreciate the reviewer’s suggestions. The naturally aged normal mice are non-genetically manipulated, on the same background as the mutant mice (B6), and maintained in the same animal facility. We have done immunostaining of CD31 and lyve1 and detected BV and lymphatics in the physiologically aged mice. We also immunostained Pax6 and found a defect in Pax6 expression as well. In addition, we have added the quantification data (Figure 6D). Vimentin and FSP1 were used to stain fibroblasts while aSMA was used to stain activated fibroblasts (myofibroblasts).

17. Results, related to Figure 7: In general, if mutant mice were aged (without wounding) would they develop the same corneal lesions (and in the same proportion of individuals) as their physiologically aged counterparts?

All the mutant mice develop corneal epithelial thinning at 2 months of age and plaque formation/metaplasia at 6 months of age, although the mice did show some variation in the severity of these phenotypes. The thinning of the epithelial layer precedes stromal remodeling and plaque formation. Our study suggest that the decreases in *Rela* expression and RA synthesis underly natural aging of the cornea. The decline in *Rela* expression is likely affected by both intrinsic and extrinsic factors during the aging process, for example, by ROS and chronic inflammation, two important pro-aging factors. We have discussed this point in the revised manuscript.

18. Results, related to Figure 8: In relation to opacity, how was this feature defined and measured. I suggest the authors identify the phenotype of the epithelium covering the cornea after RA treatment (K12/pax6/K14/ etc)

We appreciate the reviewer’s suggestions. The opacity can be affected by many factors and is not easy to define in mice. We have tried to use this word and designated corneal thinning/disruption-only mice as “moderate” and mice with corneal plaque formation as “severe”. We have added more immunostaining results in the revised manuscript, which suggest that the epithelium covering the cornea is keratinized skin-like cells (Figure 7F, Figure 7—figure supplement 1).

19. Discussion, general comment: How different is the corneal pathology in Notch mutant mice (see reference 21) compared to the defects that develop in the line being described herein?

We appreciate the reviewer’s suggestion. We have added the comparison in the discussion of the revised manuscript. There are a few differences between these two mouse models. The *Notch1* deficient mice show a loss of meibomian glands and much reduced expression of retinol transporting protein CRBP1 but not thinning or loss of the epithelial layer, compared to *K14-Cre; Rela^f/f^
*mice. Despite the differences, both models underscore the importance of RA pathway in cornea maintenance and regeneration.

20. Discussion, page 20, paragraph 2, lines 5 and 13: Does the expression of epidermal-specific genes revert to corneal-like genes after RA application? In other words, once cells have committed switch phenotype, are they able to revert back to their original lineage? Likewise, addition of VEGFR inhibitors seems to restore some of the pathological features but can this treatment restore corneal epithelial phenotype? This information is important.

The expression of epidermal-specific genes could not revert to corneal-like genes after RA application based on treatment of *K14-Cre; Rela^f/f^* mice with RA at 3 months of age, suggesting that RA cannot reverse the metaplasia process. VEGFR inhibitors could not restore corneal epithelial phenotype either. However, RA could prevent the conversion of epithelial to epidermal cells. We have clarified this issue in the revised manuscript.

21. Materials and methods, IF microscopy, IHC staining, Immunoblotting, pages 22-24: Information related to antibodies usage should be tabulated, informing the reader of the dilution and or final concentration of primary and secondary Abs used in each assay.

We appreciate the reviewer’s suggestions. We have generated a table to include all the information of all the antibodies used in this study.

22. Materials and methods, alkaline burn model, page 24: How many mice in total were used and how was this number (per group) arrived at?

We used 3-6 mice per group. We chose this number based on reported studies in the literature.

23. Materials and methods, corneal tight junction integrity, page 25: Why was this readout not quantifiable?

This readout was quantifiable and we have added quantification data.

24. Materials and methods, RA and VEGF inhibitor administration, page 25: What were the diluent controls. Were they DMSO and corn oil respectively?

For RA, the diluent control was DMSO/corn oil (1:19). For Axitinib, the diluent control was DMSO/ PEG300/Tween80/Water (1:8:1:8). The diluent was used as Veh. We have included the information in the revised manuscript.